HYPOTHESIS

# Towards a unified gating scheme for the CNBD ion channel family

Jenna L. Lin[1,2,3] and Baron Chanda[1,2,4,5]

**Cyclic nucleotide-binding domain (CNBD) channels are critical components of numerous bioelectrical processes, including cardiac pacemaking, neuronal signaling, phototransduction in the eye, and stomatal regulation in plants. While members of this channel family share a conserved overall structure, they exhibit striking differences in voltage sensitivity. Hyperpolarization-activated cyclic nucleotide-gated channels are activated by membrane hyperpolarization, whereas ether-à-go-go channels open upon depolarization. Mutagenesis and chimeragenesis studies have revealed that some mutants display bipolar gating behavior—remaining closed at intermediate membrane potentials but capable of opening in response to both hyperpolarization and depolarization. Remarkably, in certain cases, just a few mutations are sufficient to reverse the intrinsic gating polarity of the channel. This degree of diversity and plasticity in voltage-dependent gating appears to be unique to the CNBD clade and is not adequately explained by existing models. In this study, we systematically evaluate current models and propose a revised framework that better accounts for the full range of voltage-gating behaviors observed in CNBD channels.**

## Introduction

Voltage-gated ion channels (VGICs) constitute a pharmacologically important class of membrane signaling proteins that play a central role in cell signaling and sensory transduction across all kingdoms of life (Hille, 2001). Within this superfamily, cyclic nucleotide-binding domain (CNBD) channels form a distinct clade. Like other VGICs, CNBD channels share a core structure comprising six transmembrane helices; however, their distinguishing feature is the presence of an intracellular CNBD located at the C-terminal region immediately following the sixth transmembrane segment (Whicher and MacKinnon, 2016; Wang and MacKinnon, 2017; Lee and MacKinnon, 2017; Li et al., 2017; Clark et al., 2020; Li et al., 2023; Wang et al., 2025). The CNBD channels are further sorted into six subfamilies: cyclic nucleotide-gated (CNG) channels, hyperpolarization-activated cyclic nucleotide-gated (HCN) channels, ether-à-go-go (EAG) channels, plant voltage-gated K+ channels (plant VG K+ channels), plant CNG channels, and a sixth subfamily that includes channels found in prokaryotic, algal, and fungal species (Jegla et al., 2018).

The subfamilies within the CNBD family exhibit unusually diverse gating phenotypes. EAG channels, which include ether-à-go-go (Eag), Eag-related gene (Erg), and Eag-like (Elk) channels, are activated by membrane depolarization but are not sensitive to cyclic nucleotides (Drysdale et al., 1991; Warmke et al., 1991; Warmke and Ganetzky, 1994; Trudeau et al., 1995). The HCN channels are modulated by cyclic nucleotides but open upon membrane hyperpolarization, in stark contrast to every other member of the VGIC superfamily (Brown et al., 1979; Brown and Difrancesco, 1980; Ludwig et al., 1998; Gauss et al., 1998). CNG channels do not respond to changes in membrane potential and are primarily gated by cyclic nucleotides such as cAMP and cGMP (Kaupp et al., 1989; Kaupp and Seifert, 2002). The plant VG K+ channel subfamily includes both depolarization-activated (e.g., SKOR) and hyperpolarization-activated potassium ion channels (e.g., KAT1) (Jegla et al., 2018).

The first structure of a member of the CNBD family, the rat EAG channel, was solved in 2016 using single-particle cryo-EM by MacKinnon and colleagues (Whicher and MacKinnon, 2016). Since then, at least one high-resolution structure representing each of the subfamilies has become available (Li et al., 2017; Wang and MacKinnon, 2017; Lee and MacKinnon, 2017; Clark et al., 2020; Li et al., 2023; Wang et al., 2025; Dickinson et al., 2022). Although there are some differences in structural details, the overall architecture is remarkably conserved within this family. They exhibit a nondomain-swapped architecture, in

[1]Department of Anesthesiology, Washington University in St. Louis School of Medicine, St. Louis, MO, USA;   [2]Center for the Investigation of Membrane Excitability Diseases, Washington University in St. Louis School of Medicine, St. Louis, MO, USA;   [3]Graduate Program in Biochemistry, Biophysics, & Structural Biology, Washington University in St. Louis School of Medicine, St. Louis, MO, USA;   [4]Department of Biochemistry and Molecular Biophysics, Washington University in St. Louis School of Medicine, St. Louis, MO, USA;   [5]Department of Neuroscience, Washington University in St. Louis School of Medicine, St. Louis, MO, USA.

Correspondence to Baron Chanda: bchanda@wustl.edu

J.L. Lin's current affiliation is Department of Cell Biology, Harvard Medical School, Boston, MA, USA.

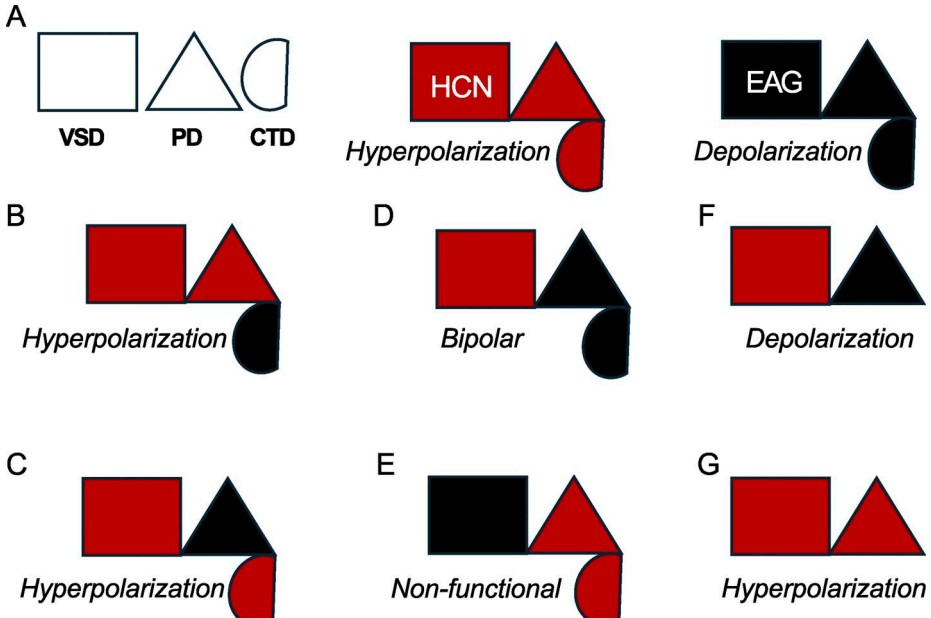

**Figure 1.** **Phenotype chart of HCN-EAG chimeras illustrates the design principles for hyperpolarization-dependent gating. (A)** Schematic showing the three structural modules that contribute to hyperpolarization-dependent gating (left). Simplified representations of HCN1 (center) and EAG (right) highlighting the corresponding modules. **(B–G)** Gating phenotypes of various chimeras and mutants. (B) HHHHE, (C) HHHEH, (D) HHHEE, (E) EEEHH, described in Cowgill et al. (2019); (F) HHHEΔC, described in Lin et al. (2024); and (G) HCN1 C-terminal deletion, from (Wainger et al, 2001) and (Wang et al, 2001). This phenotype chart shows that the HCN-derived VSD is necessary but not sufficient for hyperpolarization-dependent gating. In addition, at least one of the two secondary structural modules—either the PD or the CTD—must also be derived from HCN to confer this gating behavior. When all three modules are appropriately matched, as in wild-type HCN channels, the system exhibits more robust hyperpolarization-dependent gating. VSD, voltage-sensing domain; PD, pore domain; CTD, C-terminal domain.

which the voltage sensor is juxtaposed to the pore domain of the same subunit. These channels notably lack the S4–S5 linker helix that is characteristic of domain-swapped ion channels, such as the canonical sodium, potassium, and calcium ion channel families (Zheng and Trudeau, 2023). Instead of the S4–S5 linker helix, the S4 and S5 transmembrane segments in the CNBD channels are connected by a short unstructured (3–4 amino acid) linker.

This class of ion channels also displays remarkable functional plasticity, easily switching their gating polarity with minimal structural modifications. For example, Prole and Yellen (2006) show that cross-linking a site in the S4–S5 linker with one in the cytosolic C-linker converts the hyperpolarization-activated sea urchin HCN channel into a depolarization-activated channel. Similarly, Ramentol et al. (2020) demonstrate that the gating polarity of this channel can also be reversed by simply introducing two mutations. Additionally, single-point mutations in human ether-à-go-go–related gene (hERG) (Sanguinetti and Xu, 1999; Tristani-Firouzi et al., 2002) have been found to make the channel bipolar—it is nominally closed at intermediate potentials but opens upon hyperpolarization and depolarization. Chimeras between the various structural modules of EAG and HCN channels exhibit a range of bipolar gating phenotypes (Cowgill et al., 2019; Lin et al., 2024), which can be best conceptualized in the framework of a three-component system as shown in Fig. 1. Such diversity in gating polarity is not observed in other members of the VGIC superfamily.

Two broad classes of gating models are utilized to describe the diverse gating polarity phenotypes observed in CNBD channels. The inverted coupling model is derived from the Monod–Wyman–Changeux (Monod et al., 1965) allosteric framework initially developed for pentameric ligand-gated ion channels (Karlin, 1967; Changeux and Edelstein, 1998; Colquhoun and Lape, 2012; Auerbach, 2012; Grosman and Auerbach, 2000, 2001) and BK channels (Cox et al., 1997; Horrigan et al., 1999; Horrigan and Aldrich, 1999; Rothberg and Magleby, 2000). In this adapted version, the RCK domain from the BK channel model is replaced by the CNBD (Craven and Zagotta, 2006), with a coupling term that describes the allosteric interaction between the voltage sensor and the pore gate as the key parameter. Adjusting this single parameter enables the model to effectively describe voltage-dependent gating for either hyperpolarization- or depolarization-activated ion channels. An alternative approach is the bipolar gating model first proposed by Tristani-Firouzi and colleagues (Tristani-Firouzi et al., 2002) and later modified by (Cowgill et al, 2019). In this scheme the same channel can open in response to both hyperpolarization and depolarization stimuli, though at distinct, nonoverlapping voltage ranges. Depending on the specific model parameters, only one gating pathway is accessible in wild-type HCN and EAG channels. In this study, we systematically evaluate both classes of gating models by assessing their ability to fit the functional data reported in recent literature. We find that the inverted coupling model is cannot adequately describe the range of bipolar phenotypes observed in many mutants and HCN-EAG chimeric ion channels. Our analysis indicates that when we consider the structural constraints, a seven-state linear gating scheme with two open states can describe the observed voltage-dependent conductance of channels in this family. This model-building

exercise also provides new insights into the molecular mechanisms that determine gating polarity in this family of VGICs.

## Materials and methods

### Model fitting

All parameter values were determined by minimizing the residual sum of squares between sample data and model using Excel Solver. In all cases, the data points shown are relative open probabilities from published studies that have been normalized to the maximum current. Error bars for the data points are not shown because they cannot be extracted from the published plots. While we have access to the raw data for most of the chimeras, we have not included error bars to maintain consistency. Data and models were plotted using MATLAB R2024b.

Determination of the parameters was solved with the following conditions: $q_1$ is constrained between $[-1, -4]$, $q_2$ is constrained between $[1, 4]$, and all remaining parameters are set as nonnegative values. Due to the reference state being set in the center of all bipolar gated models, $q_1$ is set to be negatively charged to account for the bipolarity of the model. In principle, if we shift the reference state to where the voltage sensor is upon membrane hyperpolarization, we can generate the same behavior but with all positive charges. However, as a mathematical workaround to drive the voltage sensor beyond the reference state, a negative sign was assigned to the charges. The choice of reference state is entirely arbitrary and does not change the fundamental properties of the system. $x$, $K = K^0 \exp\left(\frac{-qFV}{RT}\right)$.

### Inverted coupling model
#### Equations

The gating scheme for the inverted coupling model is described in Fig. 2 A. $K_1$ is determined by Eq. 1, where $K_1^0$ is the equilibrium constant at 0 mV, and $q_1$ is the charge of the state transition.

$$K_1 = K_1^0 \exp\left(\frac{q_1 FV}{RT}\right) \tag{1}$$

where $F$ is the Faraday constant (96485 J/mol.V), $V$ is membrane potential, $R$ is the Universal gas constant (8.314 J/K.mol), and $T$ is temperature (295 K).

The open probability, $P_O$, is calculated with the following equation where $P_{OH}$ is the open probability at membrane hyperpolarization, and $P_{OD}$ is the open probability at membrane depolarization. The reference state is set as $CV_D = 1$.

$$P_{OH} = \frac{nK_1K_2}{1 + K_1 + nK_1K_2 + K_2} \tag{2}$$

$$P_{OD} = \frac{K_2}{1 + K_1 + nK_1K_2 + K_2} \tag{3}$$

$$\therefore P_O = \frac{nK_1K_2 + K_2}{1 + K_1 + nK_1K_2 + K_2} \tag{4}$$

### Three-state gating polarity model
#### Equations

The three-state gating polarity model gating scheme is shown in Fig. 3 A. In this model, $K_1$ is the same as Eq. 1, and $K_2$ is defined in the following equation:

$$K_2 = K_2^0 \exp\left(\frac{q_2 FV}{RT}\right) \tag{5}$$

The reference state is set as $C = 1$; thus, the open probability equations are such that

$$P_{OH} = \frac{K_1}{1 + K_1 + K_2} \tag{6}$$

$$P_{OD} = \frac{K_2}{1 + K_1 + K_2} \tag{7}$$

$$\therefore P_O = \frac{K_1 + K_2}{1 + K_1 + K_2} \tag{8}$$

### Five-state gating polarity model
#### Equations

The gating scheme for the five-state gating polarity model is shown in Fig. 4 A and Fig. 5 A. In this model, $K_1$ and $K_2$ are defined by Eqs. 1 and 5. $CV_R$ is set as the reference, i.e., $CV_R = 1$, resulting in calculations of the open probabilities as the following:

$$P_{OH} = \frac{K_1K_3}{K_1K_3 + K_1 + 1 + K_2 + K_2K_4} \tag{9}$$

$$P_{OD} = \frac{K_2K_4}{K_1K_3 + K_1 + 1 + K_2 + K_2K_4} \tag{10}$$

$$\therefore P_O = \frac{K_1K_3 + K_2K_4}{K_1K_3 + K_1 + 1 + K_2 + K_2K_4} \tag{11}$$

### Seven-state gating polarity model
#### Equations

For the gating scheme of the seven-state gating polarity model, see Fig. 6 A. In this model, $K_1$ and $K_2$ are defined by Eqs. 1 and 5, and $CV_R$ is set as the reference; thus, $CV_R = 1$.

The open probabilities are such that

$$P_{OH} = \frac{K_1K_3K_5}{K_1K_3K_5 + K_1K_3 + K_1 + 1 + K_2 + K_2K_4 + K_2K_4K_6} \tag{12}$$

$$P_{OD} = \frac{K_2K_4K_6}{K_1K_3K_5 + K_1K_3 + K_1 + 1 + K_2 + K_2K_4 + K_2K_4K_6} \tag{13}$$

$$\therefore P_O = \frac{K_1K_3K_5 + K_2K_4K_6}{K_1K_3K_5 + K_1K_3 + K_1 + 1 + K_2 + K_2K_4 + K_2K_4K_6} \tag{14}$$

### Six-state allosteric model
#### Equations

The gating scheme of the six-state allosteric model is shown in Fig. S1 A. Voltage-dependent steps are in the same format as Eqs. 1 and 5 with changes in respective subscripts to denote the allosteric model such that $K_A$ and $K_B$ are the following:

$$K_A = K_A^0 \exp\left(\frac{q_A FV}{RT}\right) \tag{15}$$

$$K_B = K_B^0 \exp\left(\frac{q_B FV}{RT}\right) \tag{16}$$

The following open probabilities for the six-state allosteric model uses the following equations:

$$P_{OH} = \frac{L\alpha K_A}{K_B + 1 + K_A + L\beta K_B + L + L\alpha K_A} \tag{17}$$

$$P_{OD} = \frac{L\beta K_B}{K_B + 1 + K_A + L\beta K_B + L + L\alpha K_A} \tag{18}$$

**A**

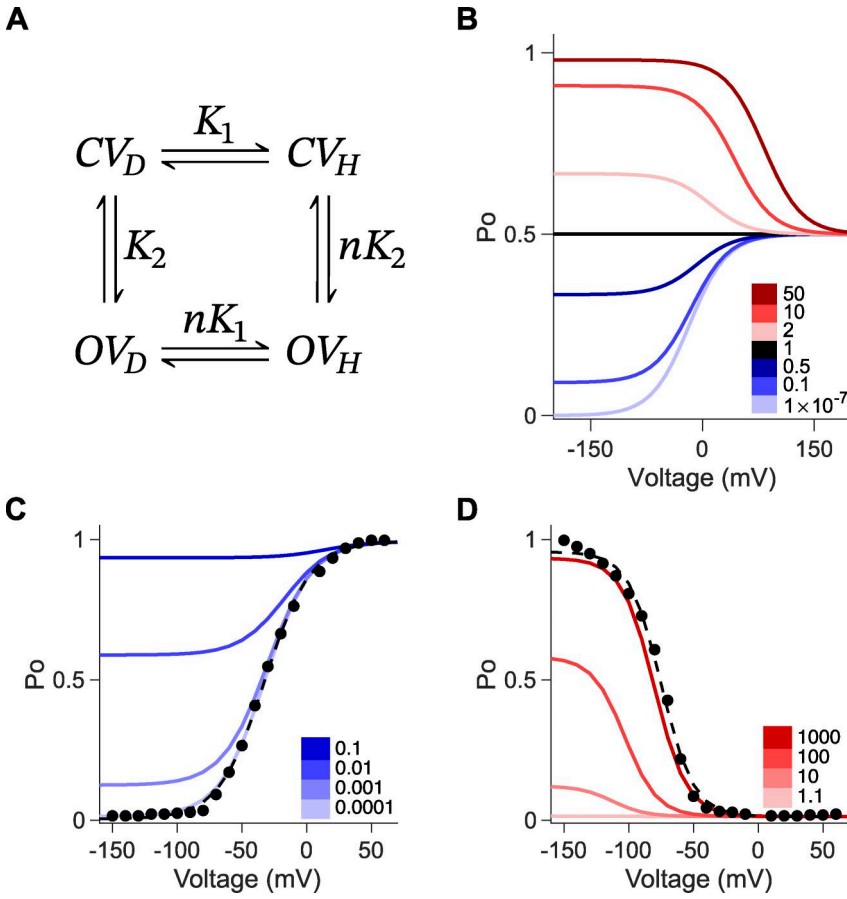

**B**

**Figure 2. Allosteric factor, *n*, of the inverted coupling model modulates gating polarity. (A)** Gating scheme of the inverted coupling model. $C$ and $O$ are when the pore is closed and opened, respectively. $V_D$ and $V_H$ are the voltage sensor upon membrane depolarization and upon membrane hyperpolarization, respectively. $K_1$ is a voltage-dependent equilibrium state constant for movement of the voltage sensor. $K_2$ is voltage-independent for opening and closure of the pore. $n$ is the allosteric factor. Equations for calculating the $P_O$–$V$ curves are described in Materials and methods. **(B)** Family of $P_O$–$V$ plots based on the gating scheme in A. The $K_1^0 = 1$, $q_1 = -1$, and $K_2 = 1$ are the same for all except for $n$, which is varied as shown in the legend. **(C)** $P_O$–$V$ scatter plot of hEAG (adapted from Cowgill et al. [2019]). These data were fitted (dashed black line) with the following parameters: $K_1^0 = 22.4$, $q_1 = -1.46$, and $K_2 = 143$. $n = 3.38 \times 10^{-5}$. $P_O$–$V$ plots in blue correspond to varying values of $n$ when $n$ is <1. **(D)** $P_O$–$V$ scatter plot of mHCN1 (adapted from Cowgill et al. [2019]). These data were fitted (dashed black line) with the following parameters: $K_1^0 = 1.13 \times 10^{-4}$, $q_1 = -2.03$, and $K_2 = 0.0139$, $n = 1,550$. $P_O$-$V$ plots in red correspond to varying values of $n$ when $n$ is >1.

**C**

**D**

$$P_{OR} = \frac{L}{K_B + 1 + K_A + L\beta K_B + L + L\alpha K_A} \quad (19)$$

$$\therefore P_O = \frac{L\beta K_B + L + L\alpha K_A}{K_B + 1 + K_A + L\beta K_B + L + L\alpha K_A} \quad (20)$$

**Seven-state allosteric model**
*Equations*
The gating scheme of the seven-state allosteric model is shown in Fig. S1 C. In this model, $K_A$ and $K_B$ are the same as in the six-state allosteric model, using Eqs. 15 and 16.

The open probabilities are the following:

$$P_{OHVH} = \frac{M\gamma K_A}{N\delta K_B + N + K_B + 1 + K_A + M + M\gamma K_A} \quad (21)$$

$$P_{OHVR} = \frac{M}{N\delta K_B + N + K_B + 1 + K_A + M + M\gamma K_A} \quad (22)$$

$$P_{ODVR} = \frac{N}{N\delta K_B + N + K_B + 1 + K_A + M + M\gamma K_A} \quad (23)$$

$$P_{ODVD} = \frac{N\delta K_B}{N\delta K_B + N + K_B + 1 + K_A + M + M\gamma K_A} \quad (24)$$

$$\therefore P_O = \frac{N\delta K_B + N + M + M\gamma K_A}{N\delta K_B + N + K_B + 1 + K_A + M + M\gamma K_A} \quad (25)$$

**Online supplemental material**
Fig. S1 describes the six-state and seven-state allosteric models that were tested to fit the HHHE-X chimeras. Fig. S2 schematically depicts the various chimeras discussed in this study. Fig. S3 shows the best fits of the constrained five-state models to

normalized conductance–voltage plots of HHHEA and HHHEK chimeras. Table S1 reports parameter values used in the unconstrained five-state gating polarity model, Table S2 reports the constrained five-state gating polarity model parameter values, Table S3 reports the parameter values of the seven-state gating polarity model, Table S4 reports the six-state allosteric model parameter values, and Table S5 reports the parameter values of the seven-state allosteric model.

## Results

### Inverted coupling model
The inverted coupling model is commonly used to describe gating in hyperpolarization-activated CNBD channels (Latorre et al., 2003; Wu et al., 2024). When $K_1$ and $K_2$ are held constant, the gating polarity is determined by the allosteric coupling factor, $n$. Channels open upon depolarization when $n$ is <1 and open upon hyperpolarization when $n$ is >1 (Fig. 2 B). A key strength of the inverted coupling model is that it captures both depolarization- and hyperpolarization-activated gating phenotypes observed in the CNBD channel family simply by varying the single parameter $n$. The model effectively describes the normalized conductance ($G/G_{max}$) for channels such as human EAG (Fig. 2 C) and mouse HCN1 (Fig. 2 D).

However, the model is limited to channels that display a single gating polarity or are constitutively open ($n = 1$) (Fig. 2, B–D), and it fails to capture the full spectrum of gating phenotypes

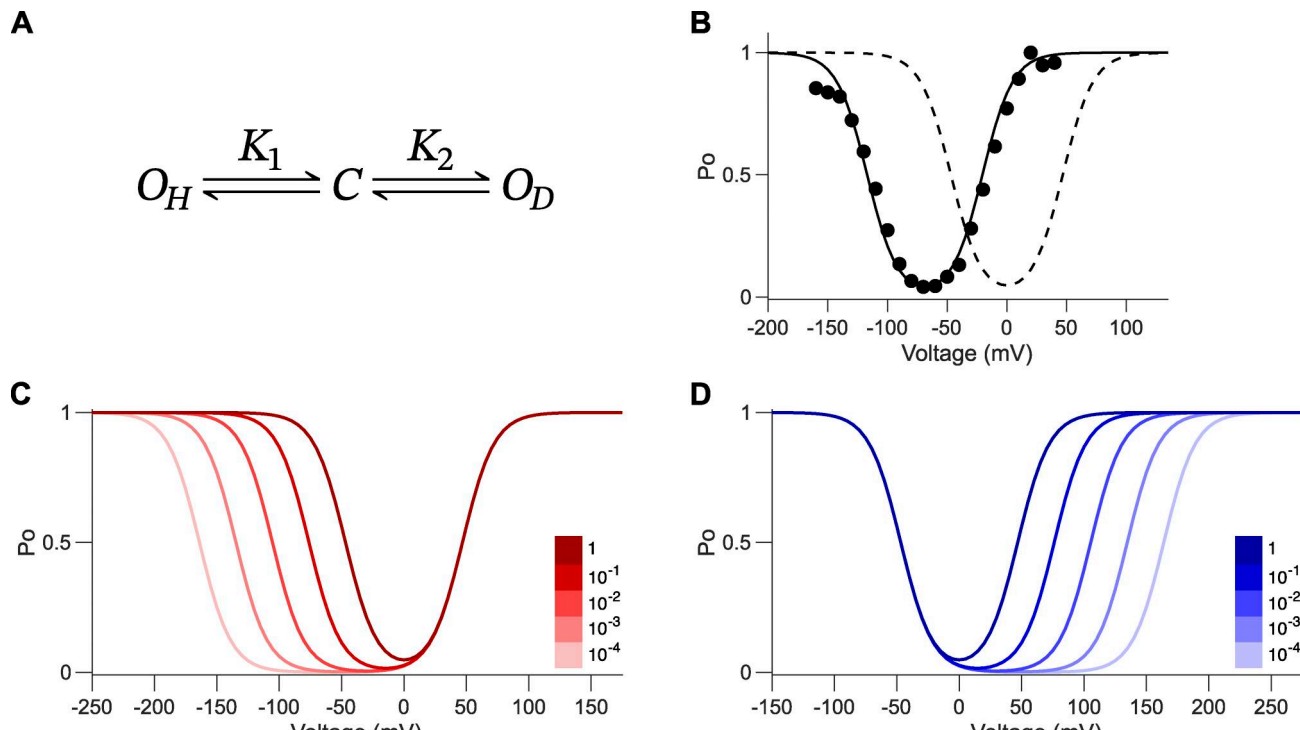

**Figure 3.** **Bipolar gating phenotype is described using the three-state gating polarity model. (A)** Gating scheme of the three-state gating polarity model. $C$ is when the channel is closed. $O_H$ is channel opening upon membrane hyperpolarization, and $O_D$ is channel opening upon depolarization. $K_1$ and $K_2$ are voltage-dependent equilibrium state constants. Equations for calculating $P_O$–$V$ curves are described in Materials and methods. All plots were generated using the parameters $q_1$ = –2 and $q_2$ = 2. **(B)** $P_O$–$V$ scatter plot of D540K-hERG mutant (adapted from Tristani-Firouzi et al. [2002]). These data were normalized again such that the maximum $P_O$ is normalized to 1 (i.e., relative current/maximum relative current). The bipolar gating phenotype is observed when $K_1^0$ = $K_2^0$ = 0.025 (dashed black line). Fitting of these data can be approximated by setting $K_1^0 < K_2^0$ such that $K_1^0$ = 0.0001 and $K_2^0$ = 5 (solid black line). **(C)** Series of $P_O$–$V$ plots show the result of $K_1^0$ becoming increasingly smaller than $K_2^0$ (solid lines with increasingly lighter shades of red), where $K_2^0$ = 0.025 and $K_1^0$ is equal to $K_2^0$ multiplied by a varying factor (shown in figure legend). **(D)** Series of $P_O$–$V$ plots show the result of $K_2^0$ becoming increasingly smaller than $K_1^0$ (solid lines with increasingly lighter shades of blue), where $K_1^0$ = 0.025 and $K_2^0$ is equal to $K_1^0$ multiplied by a varying factor (shown in figure legend).

observed in this channel family. Even when all three parameters are varied, the model cannot account for bipolar gating behavior. Although, to our knowledge, no native VGICs exhibit bipolar gating, single-point mutations in members of the CNBD family can induce this distinctive phenotype. Next, we will consider alternate models that can recapitulate bipolar gating behavior.

**Three-state gating polarity model**
One way to address the limitation of the inverted coupling model is to introduce a scheme with two distinct open states—one favored at hyperpolarized potentials and another at depolarized potentials. The simplest implementation of this idea is a three-state linear model (Fig. 3 A). This linear model was initially proposed to describe gating of the hERG channel carrying a mutation (D540K) in the S4–S5 linker of the voltage-sensing domain (VSD) (Tristani-Firouzi et al., 2002). Cowgill et al. (2019) later simplified it to a three-state model to describe gating of chimeric HCN-EAG channels.

In this scheme, the closed state ($C$) transitions to one of two open states: $O_H$, which opens upon hyperpolarization, and $O_D$, which opens upon depolarization. These transitions are governed by voltage-dependent equilibrium constants $K_1$ and $K_2$, respectively. The bipolar gating phenotype is captured by tuning the baseline values $K_1^0$ and $K_2^0$, which defines the voltage-independent component of $K_1$ and $K_2$. $K_1^0$ and $K_2^0$ are determined purely by chemical interactions, setting the midpoint. When $K_1^0$ and $K_2^0$ are equal, the model predicts a minimal open probability at 0 mV. When $K_1^0 < K_2^0$, the model shifts leftward, fitting the normalized open probability (peak tail currents, $I/I_{max}$) of D540K-hERG channels (Fig. 3 B).

How does this model account for channels with a single gating polarity, i.e., those that activate only upon hyperpolarization or depolarization? Fig. 3, C and D illustrates that adjusting $K_1^0$ or $K_2^0$ shifts the gating preference. In Fig. 3 C, lowering $K_1^0$ relative to $K_2^0$ shifts the hyperpolarization-dependent opening to more negative potentials. When $K_1^0$ becomes much smaller than $K_2^0$, the hyperpolarization gate activates so far left that the channel appears to be depolarization-gated within the experimental voltage range. Conversely, in Fig. 3 D, when $K_2^0$ is much smaller than $K_1^0$, depolarization-dependent opening is shifted far to the right, making the channel appear to activate only upon hyperpolarization.

Only under conditions where both $K_1^0$ and $K_2^0$ are sufficiently large can both gating modes be observed within an experimentally accessible voltage range. However, a key limitation of the three-state gating polarity model is that it cannot account for

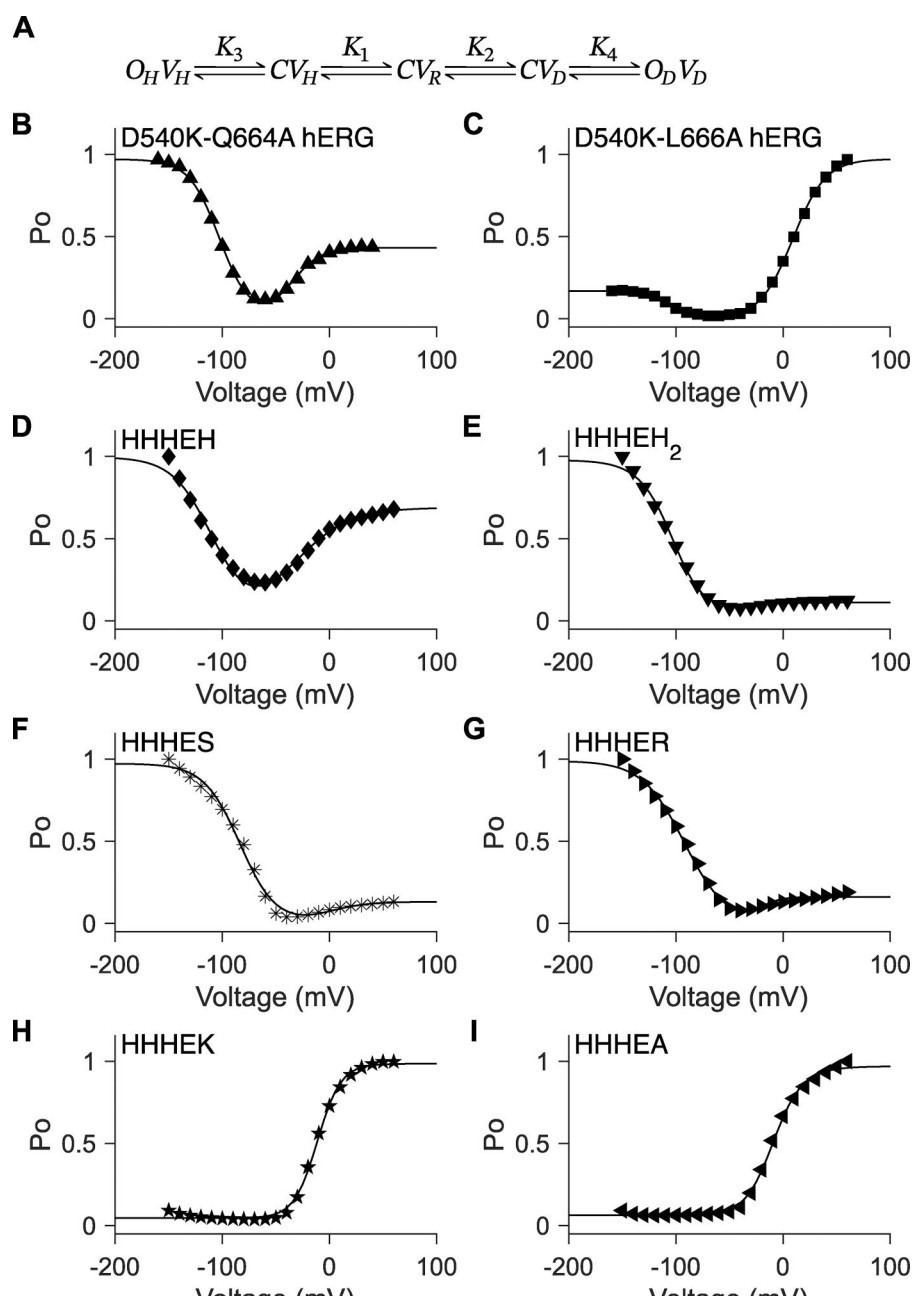

Figure 4. **Unconstrained five-state gating polarity model. (A)** Gating scheme of the five-state gating polarity model. Voltage-dependent transition steps, $K_1$ and $K_2$, and voltage-independent transition steps, $K_3$ and $K_4$, are all freely floating parameters. $C$ is the closed state, and $O$ is the open state of the pore. $V$ is the state of the voltage sensor, where $V_R$ is at rest. Subscripts $H$ and $D$ for all $O$ and $V$ states indicate states upon membrane hyperpolarization and upon membrane depolarization, respectively. Equations for calculating $P_O$–$V$ plots are described in Materials and methods. Fitting of the $P_O$–$V$ plots to data using the gating scheme are shown as a solid black line in figure panels **(B–I)**. Parameter values used for these fittings are reported in Table S1. $P_O$–$V$ scatter plot data in B and C are adapted from Tristani-Firouzi et al. (2002), and normalized again such that the maximum $P_O$ is normalized to 1 (i.e., relative current/maximum relative current). $P_O$–$V$ scatter plot data in D–I are adapted from Lin et al. (2024). **(B–I)** $P_O$–$V$ scatter plot data of D540K-Q664A hERG (upward-pointing triangle ▲), D540K-L666A hERG (square ■), HHHEH (diamond ⌣), HHHEH₂ (downward-pointing triangle ▼), HHHES (asterisk *), HHHER (right-pointing triangle ►), HHHEK (pentagram ★), and HHHEA (left-pointing triangle ◊).

differences in the maximum open probability between the two gating modes. This model always predicts that the maximum open probability approaches 1 for both states.

**Five-state gating polarity model**
To address the limitations of the three-state model, an additional closed state is introduced along both the hyperpolarizing and depolarizing pathways, rendering the final pore-opening transition voltage-independent. This results in a five-state gating polarity model (Fig. 4 A). In this model, both the VSD and pore can adopt multiple conformations, and their associated parameter values are allowed to float freely. As in the three-state model, the reference state is the central closed state ($C$), where the VSD is at rest ($V_R$). Transitions to closed states with the VSD

in the down ($V_H$) or up ($V_D$) conformation occur via voltage-dependent steps $K_1$ and $K_2$, respectively. These are followed by voltage-independent transitions $K_3$ and $K_4$ to open states at hyperpolarized ($O_H$) and depolarized ($O_D$) potentials, respectively.

Although definitive experimental evidence demonstrating that the pore-opening transition in CNBD family is entirely voltage-independent is lacking, we have treated it as such primarily to minimize the number of free parameters in our model (see also Hummert et al. [2018]). Any residual voltage dependence, if present, is expected to be minimal. Studies on Shaker potassium channels indicate that the voltage dependence associated with the pore-opening transition contributes only about 15% of the total charge movement linked to channel activation (Ledwell and Aldrich, 1999).

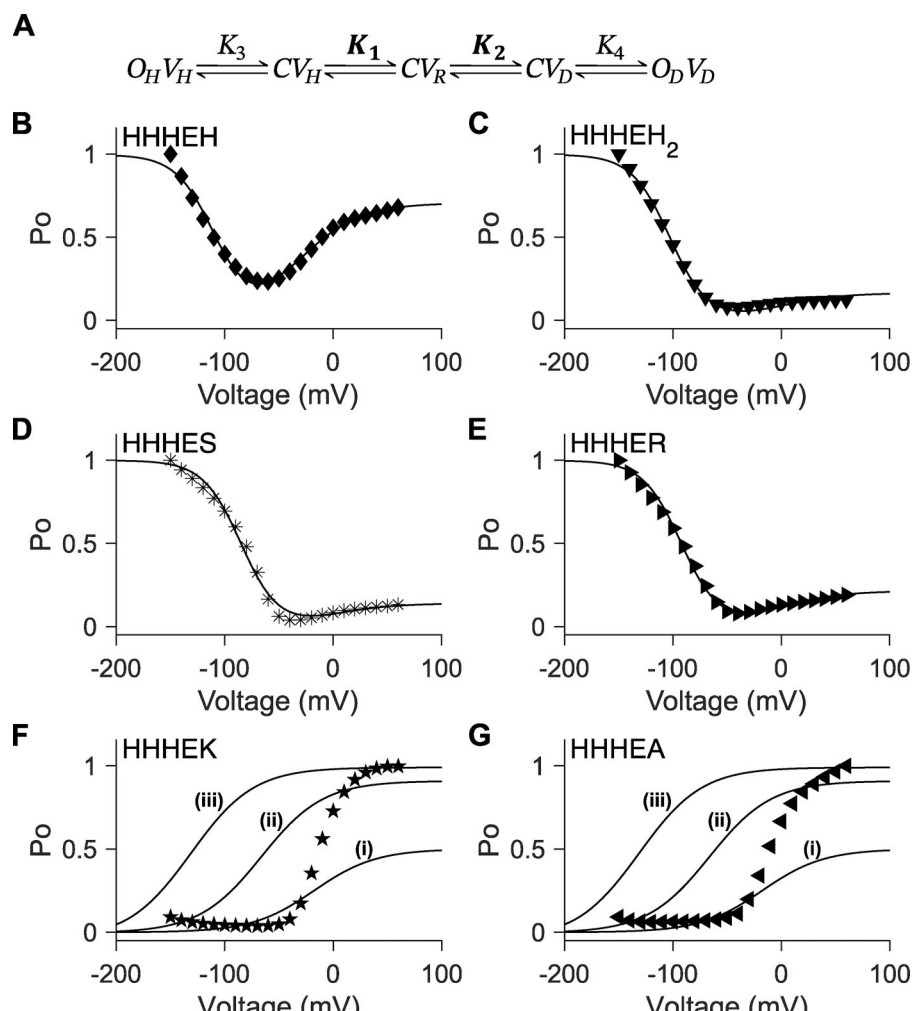

Figure 5. **Five-state gating polarity model does not fit conditions where VSD is constant. (A)** Gating scheme of the five-state gating polarity model. Voltage-dependent steps, $K_1$ and $K_2$, are bolded to indicate constrained values (i.e., $K_1$ and $K_2$ parameter values used for fitting HHHEH are applied to all HHHE-X chimeras). $K_3$ and $K_4$ are voltage-independent transition steps and vary in each chimera. Abbreviations for each state and equations for calculations of open probabilities are the same as Fig. 4 A. Fitting of $P_O$–$V$ scatter plot data to the gating scheme described in A is shown in the following panels as a solid black line. Parameter values used in these fittings are in Table S2. $P_O$–$V$ scatter plot data are adapted from Lin et al. (2024). **(B–G)** $P_O$–$V$ scatter plot of HHHEH (diamond ◡), HHHEH₂ (downward-pointing triangle ▼), HHHES (asterisk *), HHHER (right-pointing triangle ▶), HHHEK (pentagram ★), and HHHEA (left-pointing triangle ◊). **(i–iii)** Collection of $P_O$–$V$ plots in F and G, such that $K_3 = 1 \times 10^{-7}$ and $K_4$ is varied to the following parameter values: (i) $K_4 = 1$, (ii) $K_4 = 10$, and (iii) $K_4 = 100$.

With freely floating parameters, this model successfully describes the gating behavior of the D540K-Q664A and D540K-L666A hERG mutants (Tristani-Firouzi et al., 2002) (Fig. 4, B and C), which carry mutations in both the VSD and the pore domain. It can also be applied to the HCN-EAG chimeras, HHHE-X, in which the VSD is from mHCN1, the pore domain is from hEAG, and the cytosolic C terminus is replaced with elements from various CNBD family members (Lin et al., 2024) (Fig. 4, D–I).

In the HHHE-X chimeras, the VSD and pore domains remain the same, while the cytosolic C-terminal domain—comprising the C-linker and CNBD—varies (Figs. S2). While this region may influence the relative stability of the pore and, in particular, VSD conformational states, it is unlikely to directly determine the voltage dependence of VSD movement. Thus, given the conserved transmembrane domains across the HHHE-X constructs, we apply a constraint in which the voltage-dependent transitions ($K_1$ and $K_2$) are fixed to values derived from the most bipolar HHHE-X construct—HHHEH—which exhibits high relative open probability at both hyperpolarized and depolarized potentials. This assumption reflects the idea that voltage-dependent steps should be uniform across all chimeras with identical VSD and pore domains.

The constrained five-state gating polarity model (Fig. 5 A) effectively captures the gating behavior of HHHE-X constructs that open preferentially upon hyperpolarization, consistent with the behavior of HHHEH (Fig. 5, B–E). However, the model fails to describe phenotypes where depolarization-dependent opening predominates (Fig. S3). While it can produce depolarization-activated gating, the limitations of the constrained five-state model become apparent in Fig. 5, F and G. Our simulations show that $K_4$ influences both the voltage of half-maximal activation ($V_{50}$) and the maximum open probability, indicating that voltage-independent parameters simultaneously affect both the activation range and the efficacy of channel opening.

**Seven-state gating polarity model**
One approach to decouple changes in maximum open probability from shifts in $V_{50}$ is to introduce an additional voltage-independent transition along both the hyperpolarization and depolarization pathways. This yields a seven-state gating polarity model (Fig. 6 A). As with the three- and five-state models, the reference state in the seven-state model is a closed pore (C) with the VSD at rest ($V_R$). Voltage-dependent transitions $K_1$ and $K_2$ shift the voltage sensor to the down-conformation upon hyperpolarization ($V_H$) and to the up-conformation upon depolarization ($V_D$), respectively. The subsequent transitions—$K_3$, $K_4$, $K_5$, and $K_6$—are voltage-independent and define the two gating pathways.

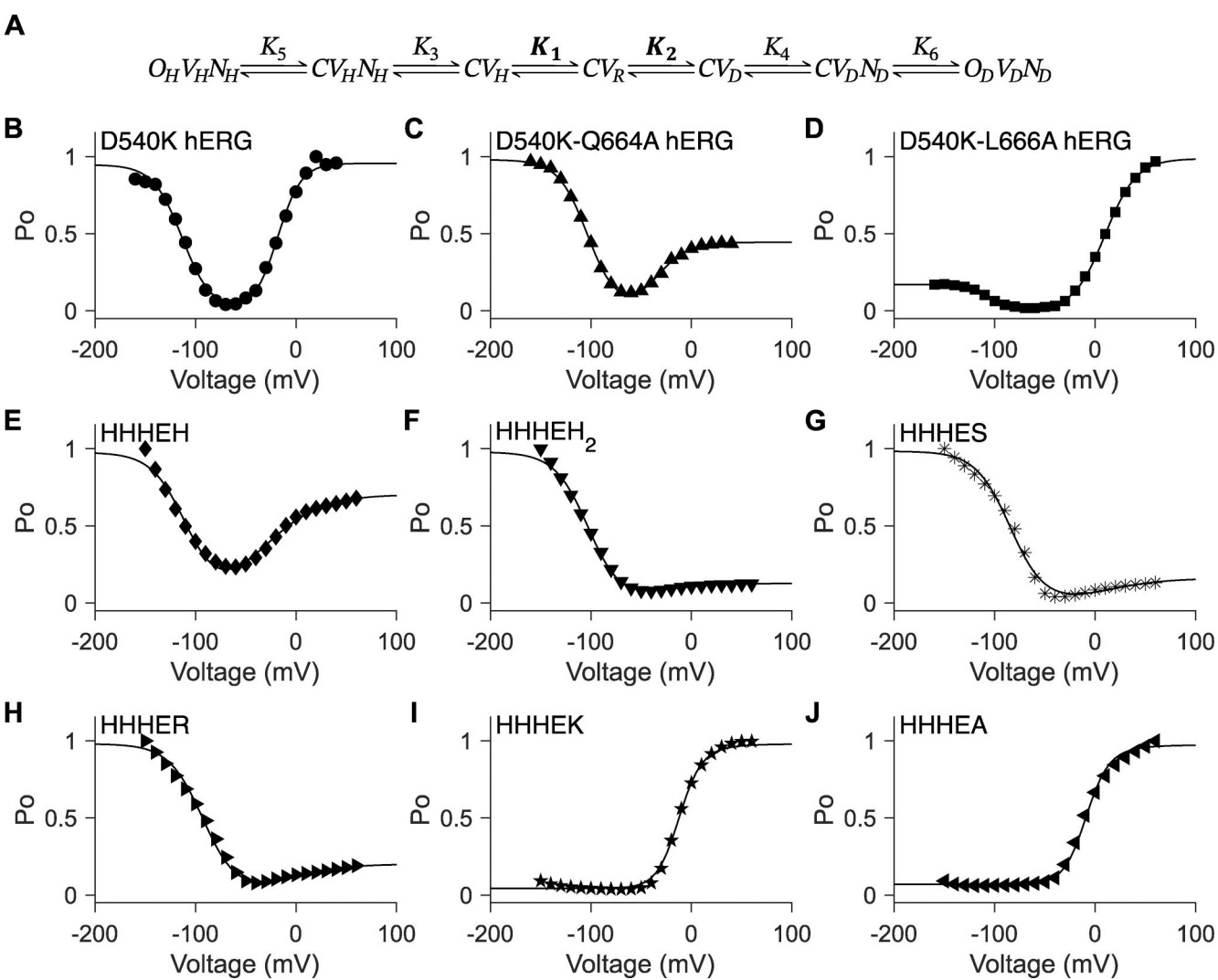

Figure 6. **Seven-state gating polarity model describes bipolar gating phenotype of HHHE-X chimeras. (A)** Gating scheme of the seven-state gating polarity model. Voltage-dependent steps, $K_1$ and $K_2$, are bolded to indicate they are constrained, where $K_1$ and $K_2$ values from HHHEH fitting are applied to all HHHE-X chimeras. $K_3$, $K_4$, $K_5$, and $K_6$ are voltage-independent transition steps and vary for each chimera. C, $O_H$, and $O_D$ are the closed and opened states of the pore, where $O_H$ is for hyperpolarization and $O_D$ is for depolarization. V is the state of the voltage sensor such that $V_R$ is at rest, $V_H$ is upon membrane hyperpolarization, and $V_D$ is upon membrane depolarization. N is the interaction due to the cytosolic C terminus upon hyperpolarization, $N_H$, and depolarization, $N_D$. Equations for open probability calculations are in Materials and methods, and parameter values used for fitting are reported in Table S3. **(B–J)** $P_O$–V plots of fitting the gating scheme in A to the $P_O$–V scatter plot data are shown as a solid black line in each figure panel. $P_O$–V scatter plot data from hERG mutants, i.e., B–D, are adapted from Tristani-Firouzi et al. (2002). These data have been normalized again such that the maximum $P_O$ value across all test potentials is normalized to 1 (i.e., relative current/maximum relative current). $P_O$–V scatter plot data from HHHE-X, i.e., E–J, are adapted from Lin et al. (2024). $P_O$–V scatter plot data of D540K hERG (circle ●), D540K-Q664A hERG (upward-pointing triangle ▲), D540K-L666A hERG (square ■), HHHEH (diamond ⌣), HHHEH$_2$ (downward-pointing triangle ▼), HHHES (asterisk *), HHHER (right-pointing triangle ▶), HHHEK (pentagram ★), and HHHEA (left-pointing triangle ◁).

The additional transition introduces intermediate closed states: $CV_HN_H$ and $CV_DN_D$, where $N_H$ and $N_D$ represent CNBD-mediated interactions that stabilize the hyperpolarized and depolarized pathways, respectively, prior to pore opening ($O_HV_HN_H$ or $O_DV_DN_D$). When applied to hERG constructs with modified VSD and pore domains, this model fits the data well when parameters are allowed to float freely (Fig. 6, B–D). For HHHE-X chimeras, we applied the constraint that all constructs share identical values for the voltage-dependent transitions $K_1$ and $K_2$, consistent with their shared transmembrane architecture. Under these constraints, the seven-state model successfully describes the gating behavior of all HHHE-X constructs (Fig. 6, E–J).

We evaluated allosteric equivalents of the three-state and five-state models (Fig. S1, A and C) to determine whether they could better fit the chimera data compared with the seven-state linear gating scheme. However, when the voltage-dependent transitions were constrained as previously described, these alternative allosteric models failed to fit the $P_O$–V curves for all HHHE-X chimeras (Fig. S1, B and D), unlike the seven-state linear gating polarity model.

## Discussion

Our model-building exercise shows that the seven-state gating polarity model provides the most parsimonious framework to

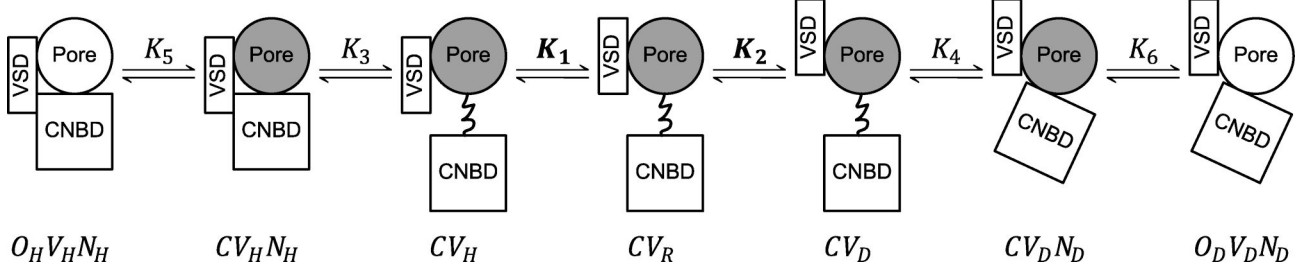

$$O_H V_H N_H \qquad CV_H N_H \qquad CV_H \qquad CV_R \qquad CV_D \qquad CV_D N_D \qquad O_D V_D N_D$$

Figure 7. **Physical interpretation of the seven-state gating polarity model for CNBD channels.** This cartoon depicts our hypothesis of the underlying mechanism in the gating scheme of Fig. 6 A. The shaded circle represents the pore in the closed state, while the blank circle is in the opened state. Distinct orientations of the CNBD are drawn to represent different interactions between the cytosolic C-terminal domain and the transmembrane domain regions.

account for the full range of voltage-dependent gating phenotypes observed in CNBD channels. This linear scheme includes two distinct open states and five closed states, reflecting the functional complexity of channels activated by both hyperpolarization and depolarization. Notably, the model interpretation (Fig. 7) posits that voltage sensor movement is followed by entry into another intermediate closed state, $CV_H N_H$ and $CV_D N_D$. These two states arise as a result of interactions between the cytosolic C-terminal domain and the transmembrane core, and these interactions are different depending on the position of the voltage sensor. This model is consistent with our findings that gating polarity of HHHE-X chimeras can be dramatically altered by either introducing C-terminal domain from other CNBD families or by making single-point mutations in the C-linker region (Cowgill et al., 2019; Lin et al., 2024).

Cryo-EM morphs of EAG channels transitioning from closed to open reveal rotational movement of the C terminus relative to the transmembrane domain during depolarization gating (Mandala and MacKinnon, 2022), while HCN1 structures across closed and open states show such movement to be minimal (Lee and MacKinnon, 2017; Burtscher et al., 2024). These structural differences support the idea that $CV_H N_H$ and $CV_D N_D$ represent distinct mechanistic modes of C-terminal interaction with the VSD and pore. Furthermore, the binding of cyclic nucleotides to the CNBD promotes C-terminal tetramerization in HCN channels, thereby relieving pore inhibition (Lolicato et al., 2011). This raises the possibility that transitions associated with $CV_H N_H$ and $CV_D N_D$ involve not only conformational changes but also oligomeric rearrangement of the cytosolic domain.

The role of the C-terminal domain in establishing gating polarity remains controversial, primarily because its deletion does not appear to alter the polarity of CNBD channels (Wang et al., 2001). We argue that this can be reconciled by considering that gating polarity arises from interactions among multiple structural modules. Analysis of HCN–EAG chimeras suggests a three-component system, in which only two components are required to establish voltage-dependent gating (Fig. 1). While the VSD is essential, it is not sufficient on its own; it must couple with either the pore domain or the CNBD to set gating polarity. Thus, the deletion of the C-terminal domain in wild-type HCN channels does not disrupt gating polarity because the pore domain can functionally substitute, highlighting redundancy in the system.

We also note that the proposal that CNBD voltage sensors access three distinct conformational states remains speculative. Although structures of EAG and HCN channels have been solved in hyperpolarized and depolarized states, it is unclear whether all three voltage sensor conformations ($V_H$, $V_D$, and $V_R$) are accessible. Hummert et al. (2018) proposed that the HCN channel voltage sensor undergoes a two-step transition, followed by a voltage-independent pore-opening transition, primarily based on modeling ionic currents. Studies by Larsson and colleagues demonstrate that the voltage sensors of spHCN channels undergo a two-step process, with the second transition of the S4 segment being associated with pore opening (Wu et al., 2021). Structural and functional studies of HCN channels suggest that the extra helical length of S4 HCN is critical for maintaining the closed state at depolarized potentials (Lee and MacKinnon, 2017; Lee and MacKinnon, 2019; Burtscher et al., 2024). The ends of the S4 and S5 helices, located near the S4–S5 linker, undergo a melting transition upon hyperpolarization, thereby relieving the pore inhibition and enabling the channels to open. There is also evidence that this region is involved in mediating coupling between the VSD and CNBD in HCN channels. It is unclear whether the voltage sensor of EAG also undergoes such a two-step movement. The S4 of EAG and other depolarized-activated channels is about one to two helical turns shorter, and it does not show any evidence of unfolding at hyperpolarized potentials. Future structural and functional studies including on channels with confirmed bipolar gating phenotypes will be essential to further establish this model.

While the seven-state model captures key aspects of CNBD channel gating, it does not fully capture all observed behaviors. For example, it does not account for inactivation, such as that seen in spHCN channels (Gauss et al., 1998). Future versions of the model may need to incorporate inactivated states. Similarly, the effects of cyclic nucleotide binding and unbinding, which are known to allosterically modulate channel gating (Goulding et al., 1994; Tibbs et al., 1997; DiFrancesco, 1999; Altomare et al., 2001; Alvarez-Baron et al., 2018), are omitted from the current scheme. Accounting for these processes will surely require expanding the model or embedding it within a broader allosteric framework. Additionally, emerging evidence suggests that in hERG channels, the N-terminal PAS domain interacts with the C terminus to influence gating (Gustina and Trudeau, 2011; Gianulis et al., 2013; Codding and Trudeau, 2018; Stevens-Sostre et al., 2024). Similarly,

the HCN domain in the N-terminal region of HCN channels interacts intimately with the CNBD and is involved in coupling voltage gating with ligand binding in HCN channels (Porro et al, 2019). While these interactions may be implicitly captured within $N_H$ and $N_D$, their specific contributions remain to be tested. Thus, our proposed seven-state model and the initial parameters should be taken as a starting point for building more detailed models of specific CNBD subfamilies rather than as a definitive model to describe their gating behavior.

In summary, this study proposes a mechanistic framework that reconciles the diverse gating behaviors observed in CNBD channels. By integrating structural data with functional analyses, we outline a model that accounts for key features of voltage-dependent gating in both native and chimeric channels. We anticipate that this framework will stimulate further studies aimed at elucidating the molecular mechanisms governing CNBD channel gating.

### Data availability

The manuscript does not present any new data. It is entirely based on published data, which have been cited appropriately. Simulations were carried out using the expressions described in the Materials and methods section.

### Acknowledgments

David A. Eisner served as editor.

We would like to thank members of the Chanda Lab for their insightful inputs and discussions. We would also like to thank Jenna L. Lin's thesis committee members for the discussions that sparked this manuscript. We thank Dr. Susovan Roy Chowdhury for help with the figures in the revised version.

This work was supported by funding from the National Institutes of Health to Baron Chanda (R35 NS116850) and Jenna L. Lin. (NIH/NINDS T32NS126120, NIH/NIGMS T32 GM008293), and UW-Madison SciMed GRS.

Author contributions: Jenna L. Lin: conceptualization, data curation, formal analysis, investigation, methodology, visualization, and writing—original draft, review, and editing. Baron Chanda: conceptualization, funding acquisition, methodology, project administration, resources, supervision, visualization, and writing—original draft, review, and editing.

Disclosures: The authors declare no competing interests exist.

Submitted: 30 June 2025

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

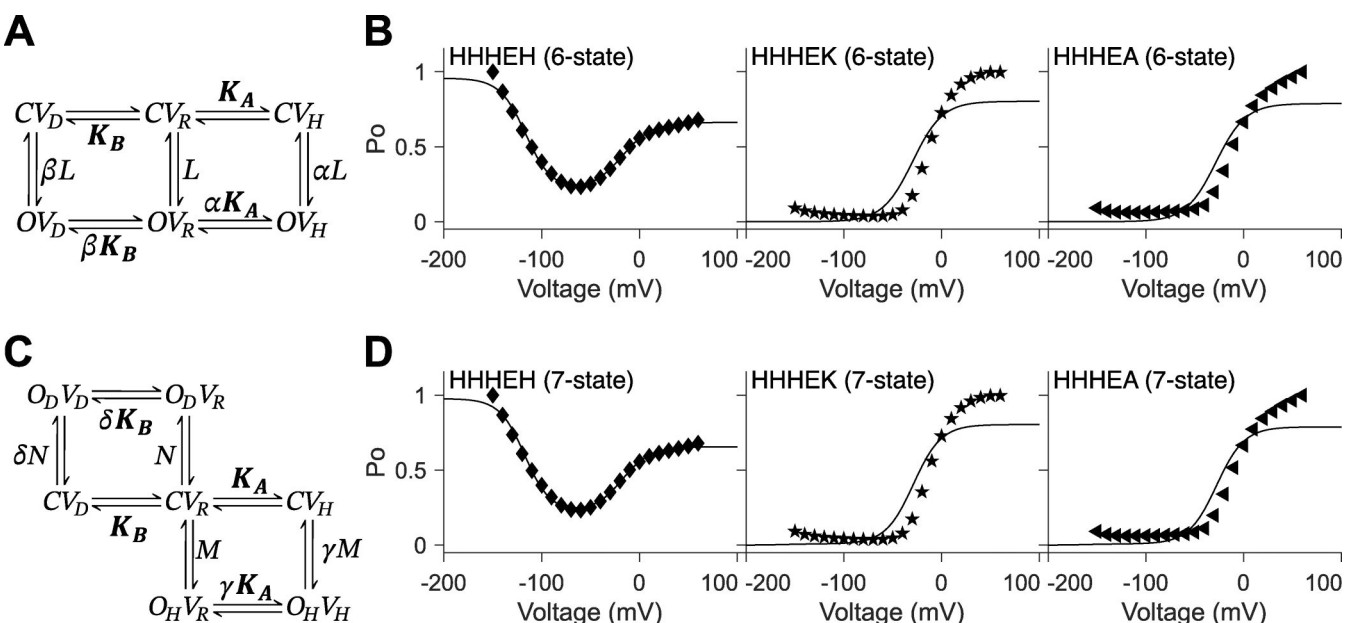

Figure S1. **Allosteric models include the five-state gating polarity model. (A and C)** $K_A$ and $K_B$ are voltage-dependent transitions; $CV_R$ is the reference (i.e., $CV_R$ = 1). $L$, $N$, and $M$ are voltage-independent transitions. $\alpha$, $\beta$, $\delta$, and $\gamma$ are allosteric factors. Voltage-dependent steps, $K_A$ and $K_B$, are bolded to indicate they are constrained such that the values for $K_A$ and $K_B$ used for fitting HHHEH are used for all HHHE-X chimeras. $C$ and $O$ are the closed and open state of the pore, respectively. $V$ is the state of the voltage sensor where $V_R$ is at rest. Subscripts $H$ and $D$ describe the states upon membrane hyperpolarization and depolarization, respectively. Equations for calculating $P_O$–$V$ plots are described in Materials and methods. In B and D, $P_O$–$V$ scatter plot data are adapted from Lin et al. (2024), and fittings of the $P_O$–$V$ plots to the corresponding gating schemes are shown as a solid black line. Parameters for allosteric models are reported in Tables S4 and S5. **(A)** Gating scheme of the six-state allosteric model. **(B)** $P_O$–$V$ plots fitting the six-state allosteric model to $P_O$–$V$ scatter plot data of HHHEH (diamond ∪), HHHEK (pentagram ★), and HHHEA (left-pointing triangle ◊). **(C)** Gating scheme of the seven-state allosteric model. **(D)** $P_O$–$V$ plots fitting the seven-state allosteric model to $P_O$–$V$ scatter plot data of HHHEH (diamond ∪), HHHEK (pentagram ★), and HHHEA (left-pointing triangle ◊).

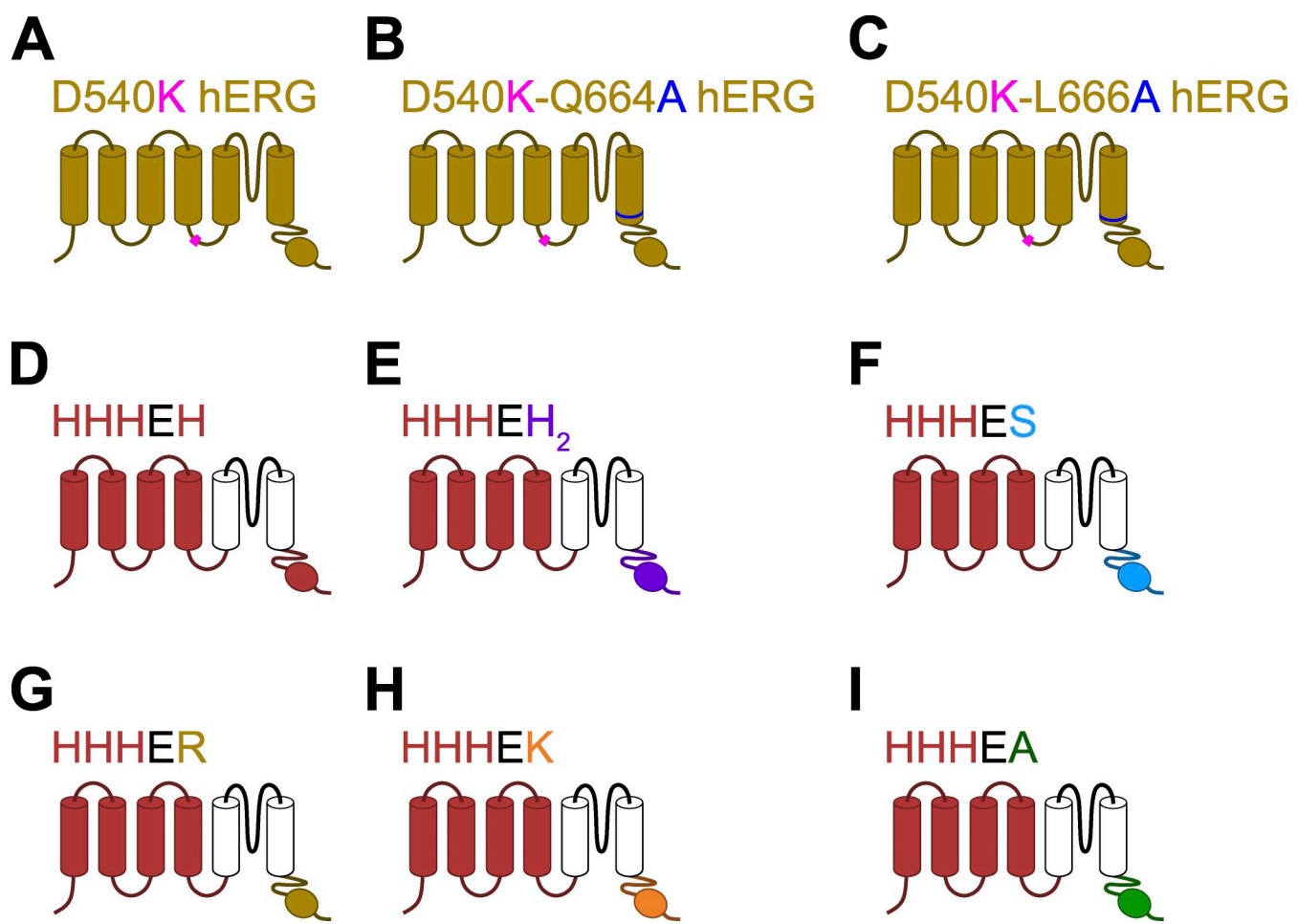

Figure S2.  **Schematic representation of the various chimeras. (A–C)** Cartoon of hERG mutants. Details of these mutants can be found in Tristani-Firouzi et al. (2002). **(D–I)** Cartoon of HHHE-X chimeras adapted from Cowgill et al. (2019); Lin et al. (2024), where the VSD is from *Mus musculus* HCN1 (shown in red) and pore is from *Homo sapiens* EAG (shown in black), while the CTD is different in each chimera. **(D)** CTD from *Mus musculus* HCN1. **(E)** CTD from *Mus musculus* HCN2. **(F)** CTD from *Strongylocentrotus purpuratus* HCN. **(G)** CTD from *Homo sapiens* ERG. **(H)** CTD from *Arabidopsis thaliana* KAT1. **(I)** CTD from *Arabidopsis thaliana* AKT1.

JGP

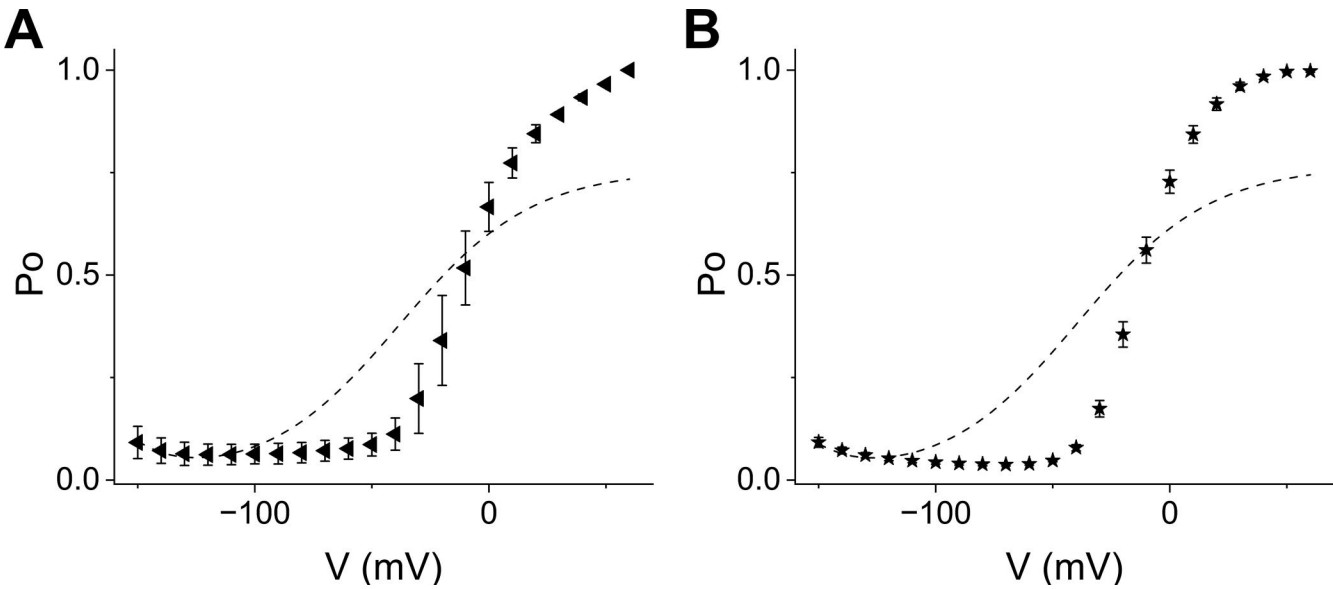

Figure S3. **Best fits of constrained five-state models to HHHEA and HHHEK. (A and B)** Normalized $P_o$-V scatter plots for HHHEA (A) (left-pointing triangle ◊) and HHHEK (B) (pentagram ★) with error bars corresponding to the standard error of means. The dashed line is the best fit obtained by minimization of the sum of squares of residuals for the constrained five-state gating polarity model. The parameter values are the same as in Table S2, except for $K_3$ and $K_4$. For HHHEK, $K_3$ and $K_4$ are 2.09 and 3.29, while for HHHEA, $K_3$ and $K_4$, are 2.30 and 3.12, respectively. The $K_3$ and $K_4$ were allowed to float freely to minimize the sum of residuals.

**Provided online are Table S1, Table S2, Table S3, Table S4, and Table S5. Table S1 shows unconstrained five-state gating polarity model parameters. Table S2 shows five-state gating polarity model parameters. Table S3 shows seven-state gating polarity model parameters. Table S4 shows six-state allosteric model parameters. Table S5 shows seven-state allosteric model parameters.**

