## [Peer Review File · The Journal of General Physiology]

Towards a unified gating scheme for the CNBD ion channel family.

Jenna Lin and Baron Chanda

Corresponding Author(s): Baron Chanda, Washington University in St. Louis School of Medicine

Review Timeline:

Submission Date:	June 30, 2025
Editorial Decision:	August 11, 2025
Revision Received:	October 9, 2025
Editorial Decision:	October 28, 2025
Revision Received:	November 18, 2025

Editor: David Eisner

Transaction Report:

DOI: <https://doi.org/10.1085/jgp.202513849>

August 11, 2025

Prof. Baron Chanda
Washington University in St. Louis School of Medicine
Department of Anesthesiology
660 S Euclid, Campus Box 8054
Rm 9305 BJCIH
St. Louis, MO 63110

Re: 202513849

Dear Baron,

Thank you for submitting your manuscript, entitled "Towards a unified gating scheme for the CNBD ion channel family." to JGP. Your manuscript has now been seen by 3 reviewers, whose comments are appended below. You will see that the reviewers have raised several concerns that should be addressed prior to further consideration of the manuscript at JGP. In particular, please view the editor's summary directly below.

Editor's Summary

The overall concern is whether your models describe the channel behaviour in a way that is an advance on previous work. Reviewer #3 argues that your model does not fit available data and provides no mechanistic insight into potential structural underpinnings of bipolar data. S/he argues that the sequential model does not provide a good description of CNBD channel gating, and that adding more parameters will not resolve this problem; instead, the model's underlying design needs to be revised. Reviewer #2 has the following major concerns. 1) The need for a more rigorous (quantitative) comparison between the inverted coupling model (successfully used in literature for CNBD-containing channels) and the sequential seven-state linear model presented by the authors; 2) How well the two models fit data in the literature, not just the constructed EAG - HCN chimeras; 3) The need for a more accurate comparison with the available structures of CNBD-containing channels. Finally, Reviewer #3 emphasizes the need for a more rigorous and quantitative comparison among the various models.

The ms was also discussed at the editors' meeting, where similar concerns were raised.

We would be pleased to receive a suitably revised manuscript that addresses these concerns, which will be re-reviewed, most likely by some or all of the original referees. In addition, please do not hesitate to contact me (via the editorial office) if you feel that a discussion of the reviewers' and editors' comments would be helpful.

Please submit your revised manuscript via the link below along with a point-by-point letter that details your responses to the editor's summary and reviewers' comments, as well as a copy of the text with alterations highlighted (boldfaced or underlined). If the article is eventually accepted, it would include a 'revised date' as well as submitted and accepted dates. If we do not receive the revised manuscript within one year, we will regard the article as having been withdrawn. We would be willing to receive a revision of the manuscript at a later time, but the manuscript will then be treated as a new submission, with a new manuscript number.

Please pay particular attention to recent changes to our instructions to authors in the following sections: Data presentation, Blinding and randomization and Statistical analysis, under Materials and Methods, as shown here: <https://rupress.org/jgp/pages/submission-guidelines#prepare>. Re-review will be contingent on inclusion of the required information (including for data added during revision) and demonstration of the experimental reproducibility of the results. Also, to improve the reproducibility of published content, we have partnered with SciScore. Authors are prompted in eJP to copy and paste the Materials and Methods section of their manuscript for a SciScore assessment when submitting their revised manuscript. Authors are encouraged (not required) to further revise their Materials and Methods if the SciScore is below 4. More information can be found here: <https://rupress.org/jgp/pages/submission-guidelines#sciscore>

Please note, JGP requires authors to submit Source Data used to generate figures containing gels and Western blots with all revised manuscripts (when applicable). This Source Data consists of fully uncropped and unprocessed images for each gel/blot displayed in the main and supplemental figures. If your paper includes cropped gel and/or blot images, please be sure to provide one Source Data file for each figure that contains gels and/or blots along with your revised manuscript files. File names for Source Data figures should be alphanumeric without any spaces or special characters (i.e., SourceDataF#, where F# refers to the associated main figure number or SourceDataFS# for those associated with Supplementary figures). The lanes of the gels/blots should be labeled as they are in the associated figure, the place where cropping was applied should be marked (with

a box), and molecular weight/size standards should be labeled wherever possible.

Source Data files will be made available to reviewers during evaluation of revised manuscripts and, if your paper is eventually published in JGP, the files will be directly linked to specific figures in the published article.

Source Data Figures should be provided as individual PDF files (one file per figure). Authors should endeavor to retain a minimum resolution of 300 dpi or pixels per inch. Please review our instructions for export from Photoshop, Illustrator, and PowerPoint here: <https://rupress.org/jgp/pages/submission-guidelines#revised>

When revising your manuscript, please be sure it is a double-spaced MS Word file and that it includes editable tables, if appropriate.

Please submit your revised manuscript via this link:

Link Not Available

Thank you for the opportunity to consider your manuscript.

Sincerely,

David Eisner, D. Phil

On behalf of Journal of General Physiology

Journal of General Physiology's mission is to publish mechanistic and quantitative molecular and cellular physiology of the highest quality; to provide a best-in-class author experience; and to nurture future generations of independent researchers.

Reviewer #1 (Comments to the Authors):

The manuscript by Lin and Chanda is aimed at providing a generalized framework to describe voltage-dependent gating of CNBD channels. CNBD channels exhibit a range of voltage-dependent gating phenotypes, including depolarization-activated (EAG), hyperpolarization-activated (HCN), and "bipolar" gating in the case of mutated and chimeric forms of the channels. The authors provide a compelling argument that it is useful to have a generalized scheme that can be applied to describe and analyze the gating of all of these channels, as it will facilitate understanding of the underlying mechanisms. To achieve the goal, the authors use a "bottom-up" approach, i.e. starting with the simplest possible gating scheme and adding on until arriving at a very simplified model that can account for the major features of voltage-dependent gating of the channels and chimeras. This is no less valid than a "top-down" approach that starts with a model based on the structure of a tetrameric channel and could account parametrically for gating motions of all of the subunit modules, which may exhibit some degree of independence. The authors show that a 7-state model can account for bipolar gating phenotypes of CNBD channel chimeras, while a 5 state model is insufficient. The authors propose a working hypothesis for a physical mechanism underlying the gating scheme in which voltage sensors move in two step, with additional states arising from voltage-independent movements of the CNBD and pore modules.

I think the manuscript in general is well written and will be of interest to readers of the JGP. I have just a couple of concerns that should be addressed.

1) When comparing models, I think it's important to provide some sort of error estimates and goodness-of-fit statistics, otherwise the conclusions are bound to be subjective to some extent. This information should be provided. Also, although the error bars have been omitted from the data for clarity, do the authors account for errors (i.e. variance) in the data when estimating the model parameters?

2) Because the authors opt for the bottom-up approach, it's not entirely clear why the jump from a 5-state to a 7-state model without demonstrating that a 6-state model is insufficient. The authors do test a 6-state allosteric model, but this is not the same as a 6-state "polarity" model, which would be intermediate between the 5 and 7-state in terms of number of parameters.

In addition, just a few minor points:

1) For the "inverted coupling" model, the gating behavior illustrated in Fig 2 C and D is driven in part by the coupling factor n ($n < 1$ is depolarization activated, $n > 1$ is hyperpolarization activated). But this is a little misleading because the open-closed equilibrium constant (K_2) is very different between the two models. Also, n is not the only parameter that modulates gating polarity, obviously q_1 can do this as well. It would not hurt to change the figure title from "Allosteric factor, n , of the inverted coupling model modulates..." to "... can modulate...".

2) Similarly in Figure 3: it would be more accurate to state that "Bipolar gating phenotype can be minimally described ..."

3) In Table S2, the values for K_3 probably do not need to be formatted in scientific notation, they are each just 3-digit numbers.

Reviewer #2 (Comments to the Authors):

This hypothesis focused paper builds on previous work involving chimeric and mutant channels within the CNBD family, which exhibit diverse biphasic, voltage-dependent gating behaviors. These gating properties are particularly unique to this channel family, as they are highly sensitive to subtle structural perturbations. This sensitivity likely arises from the non-domain-swapped architecture of the tetrameric channels, where the voltage sensor lies in close proximity to the pore domain within the same subunit. In this paper, the authors introduce a sequential 7-state gating model, which simulates voltage-dependent features of prior chimeric channel data. In contrast, a simpler 5-state model without the two end-point voltage-independent transitions failed to achieve a comparable fit. Although these sequential models achieve a good fit, the relatively high numbers of free parameters and lack of structural insight limit the novelty and mechanistic interpretation of the models. In general, adding more free parameters without structural meaning is not a good strategy to improve modeling.

The fundamental issue is that the model indicates distinct and separate voltage sensor movements are coupled to either hyperpolarization- or depolarization-dependent gating, which does not accurately reflect structural mechanisms underlying gating of this channel family. A modular allosteric scheme consisting of voltage sensing modules, a pore module, and two conformational coupling factors between the VSD and PD, and perhaps an allosteric model would be more suitable. With the major issues listed below, I do not think the current manuscript is suitable for publication, especially given the rigor and scope of JGP.

Major issues:

1. The inverted coupling model, supported by numerous previous studies, remains the most robust framework for modeling CNBD gating. Figure 2 presents an overly simplified MWC-like model and cannot be taken as evidence against the validity of the inverted coupling model. Contrary to the manuscript's claim, the inverted coupling model should not be limited to a single gating polarity, it simply requires expansion with two coupling factors to account for the U-shaped G-V.
2. Three distinct voltage sensor conformations (CVH, CVR, and CVD) was proposed in this paper. This concept inaccurately represents CNBD gating, as it implies two distinct movements of the voltage sensor-each independently coupled to either hyperpolarization- or depolarization-dependent gating. Furthermore, there is a growing consensus in the field that gating polarity of CNBD channels might not result from altered voltage sensor movement, but rather from reversed coupling mechanisms. How does this model accommodate that possibility? Could a single voltage sensor movement be linked to two distinct voltage-dependent coupling steps? In addition, how do K1 and K2 differ? It may be useful to plot their voltage dependence alongside voltage-Po relationships, possibly as voltage-dependent activation curves of VSD activation.
3. The models lack consideration of structural mechanisms. Although adding two voltage-independent steps-expanding the 5-state model to a 7-state model-was sufficient to fit the data, this improvement lacks clear biological or structural justification. What do these two voltage-independent steps represent? Also, there are too many free parameters for the 7-state model (up to 8 free parameters, including K10 and K20).
4. The linear, sequential model provides limited insight into cooperativity and allosteric regulation-features that are critical for understanding the function of tetrameric ion channel macromolecules. It lacks structural detail and cannot account for conformational coupling between domains well. I am convinced that an allosteric model is essential in this context. Have the authors tested that or thought about incorporating cooperativity and allostery into the current model?
5. Figure 1 is misleading. For example, when the VSD is derived from HCN channels while the PD and CTD are from EAG channels, it could misleadingly suggest that bipolar gating is a consistent or inherent feature for this architecture. The illustration likely applies only to specific constructs, which are not broadly interpretable. As such, the chimera results should not be overgeneralized in such way, unless they are specifically presented with residues specified. This type of schematic does not effectively support the manuscript, and without significant improvement, I recommend removing it.

Minor issues:

1. Eag-related (Erg) should be Eag-related gene (Erg) in page 3.
2. Some key references are missing. Authors should check that major papers on CNBD gating in recent years are all cited.
3. The authors should discuss that steady-state modeling has limitations and ultimately kinetic modeling is needed.

Reviewer #3 (Comments to the Authors):

In this study, Lin and Chanda compare the two current gating models used to describe the diverse gating polarity phenotypes observed in CNBD-containing channels: The sequential linear model and the inverted coupling model. They found that a seven-state linear gating scheme with two open states can describe the range of bipolar phenotypes observed in several CNBD channel mutants, and in chimeric EAG - HCN channels.

This study proposes a revised gating model scheme that better describe the several voltage-dependent gating behaviors CNBD-containing channels reported in recent literature.

As far as I understood, two key points of the proposed gating model are 1) the voltage-independent pore-opening transitions, and 2) the CNBD-mediated transitions (interaction between the cytosolic and the transmembrane machinery) that stabilize the hyperpolarized and depolarized pathways, respectively.

As for the voltage-independent pore-opening transitions, it is worth comparing the present work with the one of hummert and colleagues (PLoS Comput Biol. 2018), where they have fit a set of multiple time courses of activation and deactivation of HCN2 channels, in the absence and in the presence of cAMP, with a Markovian state model in which the voltage-dependent activation

of HCN2 channels requires a final voltage-independent pore opening step.

Moreover, comparing the present work with that of hummert and colleagues (PLoS Comput Biol. 2018) would be also helpful for a better understanding of a limit of the present gating model: The absence of the allosteric effects of binding/unbinding of cyclic nucleotides, which are known to allosterically modulate HCN and CNG channels. Such ligand-dependent effects are, instead, considered in the model published in PLoS Comput Biol. 2018.

As for the CNBD-mediated transition, i.e., the interaction between the cytosolic and the transmembrane machinery, I found somewhat contradictory the following sentence in the introduction: "These channels notably lack the S4-S5 linker helix that is characteristic of domain-swapped ion channels, such as the canonical sodium, potassium, and calcium ion channel families (Zheng and Trudeau, 2023)". As the authors stated several times in the results and in the discussion, the S4-S5 helix linker, thought reduced compared to canonical voltage-dependent sodium, potassium, and calcium ion channels, it is present at least in HCN channels and it is crucial for their gating (Saponaro et al., 2021, 2024 and Burtscher et al., 2024). When the short helix is not present, the S4-S5 linker region is anyhow crucial for the gating of CNBD-containing channels (see Cowgill et al., 2019 and Lin et al., 2024 for EAG; Clark et al., 2020 for KAT1; Hu et al., 2023 for CNG). Please, rephrase the sentence in the introduction and clarify the concept of the S4-S5 linker as a key gating element in CNBD-containing channels.

Moreover, in HCN1 (Burtscher et al., 2024) and in HCN4 (Saponaro et al., 2021, 2024) the short helix in the S4-S5 linker undergoes unfolding during gating. Such transition is also coupled with a concomitant degree of uncoupling between S4 (VSD) and S5 (pore) (Lee and MacKinnon 2017, 2019 and Burtscher et al., 2024 for HCN1; Saponaro et al., 2021, 2024 for HCN4). I consider the above-described structural features of HCNs relevant for the discussion where the authors speculate about the multiple conformations S4 can adopt, as well as when they infer that "a complicating factor (for a fully comprehensive model of CNBD-containing channels) is the significant difference in S4 helix length between depolarization-activated and hyperpolarization-activated channels". Therefore, I suggest to introduce such elements in the discussion.

It is also worth noting that S4-S5 helix linker mediates cAMP signal transduction in the ligand sensitive HCN4 subtype (Saponaro et al., 2021). Again, this is interesting for the understanding of the CNBD-mediated transitions proposed by the present gating model and corroborates, from the structural point of views, its existence.

Moreover, HCN channels display an N-terminal folded helical domain (HCND, Lee and Mackinnon, 2017), that has been shown to be involved in the CNBD-mediated gating mechanism of HCN channels (Porro et al., 2019). Therefore, I suggest to state the role of the HCND in HCN channels' gating mechanism needs in the discussion, as the authors have been properly done for the N-terminal PAS domain of the EAG channels

Response to Reviewers:

Reviewer #1 (Comments to the Authors):

The manuscript by Lin and Chanda is aimed at providing a generalized framework to describe voltage-dependent gating of CNBD channels. CNBD channels exhibit a range of voltage-dependent gating phenotypes, including depolarization-activated (EAG), hyperpolarization-activated (HCN), and "bipolar" gating in the case of mutated and chimeric forms of the channels. The authors provide a compelling argument that it is useful to have a generalized scheme that can be applied to describe and analyze the gating of all of these channels, as it will facilitate understanding of the underlying mechanisms. To achieve the goal, the authors use a "bottom-up" approach, i.e. starting with the simplest possible gating scheme and adding on until arriving at a very simplified model that can account for the major features of voltage-dependent gating of the channels and chimeras. This is no less valid than a "top-down" approach that starts with a model based on the structure of a tetrameric channel and could account parametrically for gating motions of all of the subunit modules, which may exhibit some degree of independence. The authors show that a 7-state model can account for bipolar gating phenotypes of CNBD channel chimeras, while a 5 state model is insufficient. The authors propose a working hypothesis for a physical mechanism underlying the gating scheme in which voltage sensors move in two step, with additional states arising from voltage-independent movements of the CNBD and pore modules.

I think the manuscript in general is well written and will be of interest to readers of the JGP. I have just a couple of concerns that should be addressed.

We appreciate your constructive feedback and positive comments. Please see our responses below:

1) When comparing models, I think it's important to provide some sort of error estimates and goodness-of-fit statistics, otherwise the conclusions are bound to be subjective to some extent. This information should be provided. Also, although the error bars have been omitted from the data for clarity, do the authors account for errors (i.e. variance) in the data when estimating the model parameters?

The reviewer makes a good point. The main reason we cannot display error bars is that we simply do not have access to that data. In much of the published literature, the original raw data is not available, and error bars are only shown in the final figures, and sometimes "error bars are smaller than the symbol!" Therefore, we are able to reliably extract the data points from these plots but not the error values.

When the data was collected in our lab, we had access to complete datasets and could generate error bars. However, for the sake of uniformity, we did not show the error bars. We have now made this change and noted it in the Methods section (Pg. 4, line 24).

We have also highlighted that our goal is not to find the exact parameters that fit the experimental data, but rather to broadly identify the type of model that is sufficient and provide initial parameter values for a detailed model-building exercise (Pg. 12, line 24). In most cases, model adequacy was determined either by visual examination of fits or by analyzing the trends of the simulated data. Nevertheless, we have added a supplementary figure to the constrained 5-state model, showing that by simply optimizing parameters K_3 and K_4 , we are unable to fit the experimental data for HHHEK and HHHEA. (see below)

Figure S3. **Best fits of constrained five-state models to HHHEA and HHHEK.** Normalized I/I_{\max} - V scatter plots for HHHEA (A) (left-pointing triangle ◄) and HHHEK (B) (pentagram ★) with error bars corresponding to the standard error of means. The dashed line is the best fit obtained by minimization of the sum of squares of residuals for the five-state gating polarity model. The parameter values are the same as in Table S2, except for K_3 and K_4 . For HHHEK, K_3 and K_4 are 2.09 and 3.29, while for HHHEA, K_3 and K_4 are 2.30 and 3.12, respectively. The K_3 and K_4 were allowed to float freely to minimize the sum of residuals.

2) Because the authors opt for the bottom-up approach, it's not entirely clear why the jump from a 5-state to a 7-state model without demonstrating that a 6-state model is insufficient. The authors do test a 6-state allosteric model, but this is not the same as a 6-state "polarity" model, which would be intermediate between the 5 and 7-state in terms of number of parameters.

It is possible that a 6-state model would have been sufficient; however, it would have introduced asymmetry into the model and implied that the cytoplasmic domain contributes to either the hyperpolarization or depolarization pathway. There is evidence in the literature that the cytoplasmic domains modulate the gating of both the HCN and hERG channels (Pg. 12, lines 19-23).

As shown in Figs. 5F and G, modulating the final transition modifies both the voltage dependence of opening and the maximum open probability. To reduce the co-dependence of these parameters, we added an additional transition in both the hyperpolarization and depolarization-dependent pathways.

In addition, just a few minor points:

1) For the "inverted coupling" model, the gating behavior illustrated in Fig 2 C and D is driven in part by the coupling factor n ($n < 1$ is depolarization activated, $n > 1$ is hyperpolarization activated). But this is a little misleading because the open-closed equilibrium constant (K_2) is very different between the two models. Also, n is not the only parameter that modulates gating polarity, obviously q_1 can do this as well. It would not hurt to change the figure title from "Allosteric factor; n , of the inverted coupling model modulates..." to "... can modulate..."

Agreed. The title was modified.

2) Similarly in Figure 3: it would be more accurate to state that "Bipolar gating phenotype can be minimally described ..."

Modified as suggested.

3) In Table S2, the values for K3 probably do not need to be formatted in scientific notation, they are each just 3-digit numbers.

Agreed.

Reviewer #2 (Comments to the Authors):

This hypothesis focused paper builds on previous work involving chimeric and mutant channels within the CNBD family, which exhibit diverse biphasic, voltage-dependent gating behaviors. These gating properties are particularly unique to this channel family, as they are highly sensitive to subtle structural perturbations. This sensitivity likely arises from the non-domain-swapped architecture of the tetrameric channels, where the voltage sensor lies in close proximity to the pore domain within the same subunit. In this paper, the authors introduce a sequential 7-state gating model, which simulates voltage-dependent features of prior chimeric channel data. In contrast, a simpler 5-state model without the two end-point voltage-independent transitions failed to achieve a comparable fit. Although these sequential models achieve a good fit, the relatively high numbers of free parameters and lack of structural insight limit the novelty and mechanistic interpretation of the models. In general, adding more free parameters without structural meaning is not a good strategy to improve modeling.

The fundamental issue is that the model indicates distinct and separate voltage sensor movements are coupled to either hyperpolarization- or depolarization-dependent gating, which does not accurately reflect structural mechanisms underlying gating of this channel family. A modular allosteric scheme consisting of voltage sensing modules, a pore module, and two conformational coupling factors between the VSD and PD, and perhaps an allosteric model would be more suitable. With the major issues listed below, I do not think the current manuscript is suitable for publication, especially given the rigor and scope of JGP.

We thank the reviewer for their detailed feedback. Please see our responses below.

Major issues:

1. The inverted coupling model, supported by numerous previous studies, remains the most robust framework for modeling CNBD gating. Figure 2 presents an overly simplified MWC-like model and cannot be taken as evidence against the validity of the inverted coupling model. Contrary to the manuscript's claim, the inverted coupling model should not be limited to a single gating polarity, it simply requires expansion with two coupling factors to account for the U-shaped G-V.

We were wondering what the reviewer meant by adding two coupling factors since we are not aware of any allosteric model where the connection between the same two transitions is associated with two coupling factors. While this manuscript was being revised, we noticed that a preprint describing an allosteric model for the gating of CNBD channels was posted on August 1st.

<https://www.biorxiv.org/content/10.1101/2025.07.28.666606v1>

The top panel is the allosteric model with two coupling factors, and the bottom panel is our five-state model.

The allosteric model with two coupling terms is more complicated than the bipolar gating model for the following reasons:

C_{00} which corresponds to the Closed uncoupled state is C_{VR} in our model; C_{D0} corresponds to the closed state coupled to depolarization pathway which is equivalent to C_{VD} , and C_{H0} corresponds to closed state coupled to hyperpolarized pathway which is equivalent to C_{VH} .

The allosteric model has an additional C_{DH} , which is the closed state that is coupled to both the hyperpolarization and depolarization states. We have no idea what this state means physically.

This model has 5 model parameters and 8 states as opposed to four parameters and 5 states in our model. Note that the unconstrained five-state model can fit all the datasets (see Fig. 4). Additional constraints were introduced to account for the chimera data which is why we had to go the seven-state model. In contrast, the model published in Biorxiv does not take any of the structural constraints into account.

We couldn't agree more with the reviewer that "adding more free parameters without structural meaning is not a good strategy to improve modeling."!

2, Three distinct voltage sensor conformations (CVH, CVR, and CVD) was proposed in this paper. This concept inaccurately represents CNBD gating, as it implies two distinct movements of the voltage sensor, each independently coupled to either hyperpolarization- or depolarization-dependent gating.

Our hypothesis is that there are two distinct voltage-sensor movements corresponding to three states. We disagree with the reviewer that this is inaccurate or unsupported by data. VCF measurements in the spHCN channels show that the voltage sensor undergoes a two-step movement (PMID: 34504015). Similar two-step voltage-sensor conformations have been observed in other voltage-gated ion channels, most notably KCNQ1+KCNE channels. We agree that it has not been demonstrated whether these movements correspond to hyperpolarization- and depolarization-dependent gating, as no experiments have been conducted in bipolar mutants. Since this is a hypothesis paper, our goal is to connect disparate experimental data and propose a testable model. Nevertheless, we do make it clear in the discussion that this idea remains speculative (Line 31, Pg.11).

Furthermore, there is a growing consensus in the field that gating polarity of CNBD channels might not result from altered voltage sensor movement, but rather from reversed coupling mechanisms. How does this model accommodate that possibility?

We disagree with the majoritarian view, which mainly explains the unipolar gating behavior. As we have shown, the existing reversed coupling cannot account for the bipolar phenotype (Figure 2). Before the preprint, the only model that could account for bipolar gating was the one first described by Sanguinetti, as acknowledged in our manuscript (Line 21, Pg. 8).

Could a single voltage sensor movement be linked to two distinct voltage-dependent coupling steps? In addition, how do K1 and K2 differ? It may be useful to plot their voltage dependence alongside voltage-Po relationships, possibly as voltage-dependent activation curves of VSD activation.

K_1 and K_2 are the two voltage-dependent transitions. We have shown below the plots of the voltage-dependence of their values.

Trends of Seven-State Gating Polarity Model. Incremental changes made to each parameter noted on the left side of each plot. The left column in red represents P_{OH} , right column in blue is P_{OD} , and in black is P_O in both columns. Increasing parameter values are shown with increasing opacity of each line. (A) Changing parameter value for K_1^0 . (B) Changing parameter value K_2^0 .

3. The models lack consideration of structural mechanisms. Although adding two voltage-independent steps-expanding the 5-state model to a 7-state model-was sufficient to fit the data, this improvement lacks clear biological or structural justification. What do these two voltage-independent steps represent?

The 7-state model accounts for the role of cytoplasmic domains in modulating voltage-gated ion channels. We have discussed this in the following section (Line 8, Pg.11):

“Notably, the model interpretation (**Fig. 7**) posits that voltage sensor movement is followed by entry into another intermediate closed state, $CV_H N_H$ and $CV_D N_D$. These two states arise as a result of interactions between the cytosolic C-terminal domain and the transmembrane core, and these interactions are different depending on the position of the voltage-sensor. This model is consistent with our findings that gating polarity of HHHE-X chimeras can be dramatically altered by either introducing C-terminal domain from other CNBD families or by making single-point mutations in the C-linker region (Cowgill et al., 2019; Lin et al., 2024).”

This has also been shown in Figure 7, where we have positioned the CNBD differently in the hyperpolarized and depolarized states. These two interactions define the two voltage-independent transitions.

Also, there are too many free parameters for the 7-state model (up to 8 free parameters, including K10 and K20).

There are no extra parameters. Voltage-dependent transitions will always have two parameters: one corresponding to the chemical interaction and the other defining the voltage-dependence.

4. The linear, sequential model provides limited insight into cooperativity and allosteric regulation-features that are critical for understanding the function of tetrameric ion channel macromolecules. It lacks structural detail and cannot account for conformational coupling between domains well. I am convinced that an allosteric model is essential in this context. Have the authors tested that or thought about incorporating cooperativity and allostery into the current model?

We would like to highlight Zagotta, Hoshi, and Aldrich's model for the Shaker potassium channel to the reviewer. In this model, the pore domain and voltage-sensor domain are considered modular, but the coupling is not shown as an allosteric scheme. Instead, it is depicted as a straightforward sequential process. Shaker voltage-sensor activation is obligatorily linked to pore opening. This model is highly cited in the field and has provided significant insights.

We are unsure of the exact source of the confusion, but we encourage the reviewer to consult the following paper (PMID:34139217 (see the section on allosteric linkage analysis)), where the authors specifically discuss considerations for using allosteric versus sequential models.

5. Figure 1 is misleading. For example, when the VSD is derived from HCN channels while the PD and CTD are from EAG channels, it could misleadingly suggest that bipolar gating is a consistent or inherent feature for this architecture. The illustration likely applies only to specific constructs, which are not broadly interpretable. As such, the chimera results should not be overgeneralized in such way, unless they are specifically presented with residues specified. This type of schematic does not effectively support the manuscript, and without significant improvement, I recommend removing it.

Figure 1 is necessary to summarize, in cartoon form, the experimental data on how the three structural modules determine the gating polarity of these channels. This opening figure provides a context for understanding Figure 6, where we build a hypothetical model grounded on these structural elements.

Bipolar gating is observed not only in the chimeras but also in mutants of the spHCN and hERG channels. It is our fundamental belief that any general model has to not only explain the functional properties of the wild-type channels but also their derivatives. Sometimes, the derivatives are more useful for mechanistic studies than the wild-type channels. Note that the Shaker IR (inactivation removed) channel, not the wild-type Shaker, was used to study the potassium channel gating and build the ZHA model.

Minor issues:

1. Eag-related (Erg) should be Eag-related gene (Erg) in page 3.

Done

2. Some key references are missing. Authors should check that major papers on CNBD gating in recent years are all cited.

Added.

3. The authors should discuss that steady-state modeling has limitations and ultimately kinetic modeling is needed.

We have clarified that our model and its parameters should be used as a starting point for developing

detailed models (Line 24, Pg 12). The JGP readers are quite sophisticated, so it is unnecessary to include such a qualification. Considering the limitations of kinetic modeling, adding a statement without further explanation would be pointless.

Reviewer #3 (Comments to the Authors):

In this study, Lin and Chanda compare the two current gating models used to describe the diverse gating polarity phenotypes observed in CNBD-containing channels: The sequential linear model and the inverted coupling model. They found that a seven-state linear gating scheme with two open states can describe the range of bipolar phenotypes observed in several CNBD channel mutants, and in chimeric EAG - HCN channels.

This study proposes a revised gating model scheme that better describe the several voltage-dependent gating behaviors CNBD-containing channels reported in recent literature.

We thank the reviewer for their constructive comments. Please see our responses below:

As far as I understood, two key points of the proposed gating model are 1) the voltage-independent pore-opening transitions, and 2) the CNBD-mediated transitions (interaction between the cytosolic and the transmembrane machinery) that stabilize the hyperpolarized and depolarized pathways, respectively.

The other key feature of the model is that the voltage-sensing involves two voltage-dependent transitions (starting line 31, Pg. 11).

As for the voltage-independent pore-opening transitions, it is worth comparing the present work with the one of hummert and colleagues (PLoS Comput Biol. 2018), where they have fit a set of multiple time courses of activation and deactivation of HCN2 channels, in the absence and in the presence of cAMP, with a Markovian state model in which the voltage-dependent activation of HCN2 channels requires a final voltage-independent pore opening step.

Moreover, comparing the present work with that of hummert and colleagues (PLoS Comput Biol. 2018) would be also helpful for a better understanding of a limit of the present gating model: The absence of the allosteric effects of binding/unbinding of cyclic nucleotides, which are known to allosterically modulate HCN and CNG channels. Such ligand-dependent effects are, instead, considered in the model published in PLOS Computational Biology. 2018.

After it became clear that the three-state model could not, we had to separate the pore opening transitions. The simple reason for keeping it voltage-independent is that we have fewer parameters. To the best of our knowledge, there is no definitive evidence that pore opening in HCN channels is completely voltage-independent. In the Shaker potassium channel, the Aldrich lab created the ILT mutation, which decoupled the voltage-sensor transition from pore opening. In that case, they found that pore opening still exhibits residual voltage-dependence. We have clarified this in the section (line 21, Pg. 9).

As for the CNBD-mediated transition, i.e., the interaction between the cytosolic and the transmembrane machinery, I found somewhat contradictory the following sentence in the introduction: "These channels notably lack the S4-S5 linker helix that is characteristic of domain-swapped ion channels, such as the canonical sodium, potassium, and calcium ion channel families (Zheng and Trudeau, 2023)". As the

authors stated several times in the results and in the discussion, the S4-S5 helix linker; though reduced compared to canonical voltage-dependent sodium, potassium, and calcium ion channels, it is present at least in HCN channels and it is crucial for their gating (Saponaro et al., 2021, 2024 and Burtscher et al., 2024). When the short helix is not present, the S4-S5 linker region is anyhow crucial for the gating of CNBD-containing channels (see Cowgill et al., 2019 and Lin et al., 2024 for EAG; Clark et al., 2020 for KATI; Hu et al., 2023 for CNG). Please rephrase the sentence in the introduction and clarify the concept of the S4-S5 linker as a key gating element in CNBD-containing channels.

We have added an additional sentence (line 18, Pg 3):

Instead of the S4-S5 linker helix, the S4 and S5 transmembrane segments in CNBD channels is connected by a short unstructured (3-4 amino acid) linker.

Moreover, in HCN1 (Burtscher et al., 2024) and in HCN4 (Saponaro et al., 2021, 2024) the short helix in the S4-S5 linker undergoes unfolding during gating. Such transition is also coupled with a concomitant degree of uncoupling between S4 (VSD) and S5 (pore) (Lee and MacKinnon 2017, 2019 and Burtscher et al., 2024 for HCN1; Saponaro et al., 2021, 2024 for HCN4). I consider the above-described structural features of HCNs relevant for the discussion where the authors speculate about the multiple conformations S4 can adopt, as well as when they infer that "a complicating factor (for a fully comprehensive model of CNBD-containing channels) is the significant difference in S4 helix length between depolarization-activated and hyperpolarization-activated channels". Therefore, I suggest to introduce such elements in the discussion.

The reviewer makes a good point. We have eliminated that specific sentence and rewritten this section to discuss the structural and other data regarding the conformational changes in this region (see line 32, Pg. 11). Thank you for bringing this to our attention.

It is also worth noting that S4-S5 helix linker mediates cAMP signal transduction in the ligand sensitive HCN4 subtype (Saponaro et al., 2021). Again, this is interesting for the understanding of the CNBD-mediated transitions proposed by the present gating model and corroborates, from the structural point of views, its existence.

Addressed in the previous response.

Moreover, HCN channels display an N-terminal folded helical domain (HCND, Lee and Mackinnon, 2017), that has been shown to be involved in the CNBD-mediated gating mechanism of HCN channels (Porro et al., 2019). Therefore, I suggest to state the role of the HCND in HCN channels' gating mechanism needs in the discussion, as the authors have adequately been done for the N-terminal PAS domain of the EAG channels.

Agreed! We have mentioned this in the text (Line 21, pg. 12).

Prof. Baron Chanda
Washington University in St. Louis School of Medicine
Department of Anesthesiology
660 S Euclid, Campus Box 8054
Rm 9305 BJCIH
St. Louis, MO 63110

Re: 202513849R1

Dear Baron,

I am pleased to let you know that your manuscript, titled "Towards a unified gating scheme for the CNBD ion channel family." is scientifically acceptable for publication in Journal of General Physiology. Two of the reviewers have made suggestions which I encourage you to consider. Formal acceptance will follow when it is modified in accordance with our editorial policies.

Please note items that need attention are listed at the bottom of this email (under 'manuscript formatting checklist'). Please also be sure to include a letter addressing the reviewers' comments point-by-point (if applicable) and a copy of the text with alterations highlighted (boldfaced or underlined). Your manuscript should be a double-spaced MS Word file and include editable tables, if appropriate.

Lastly, JGP requires a data availability statement for all research article submissions. These statements will be published in the article directly above the Acknowledgments. The statement should address all data underlying the research presented in the manuscript. Please visit the JGP instructions for authors for guidelines and examples of statements at <https://rupress.org/jgp/pages/editorial-policies#data-availability-statement>.

Please submit your final files via this link:
Link Not Available

Thank you for choosing to publish your research in JGP and please feel free to contact me with any questions.

Sincerely,

David Eisner, D. Phil
On behalf of Journal of General Physiology

Journal of General Physiology's mission is to publish mechanistic and quantitative molecular and cellular physiology of the highest quality; to provide a best in class author experience; and to nurture future generations of independent researchers.

Manuscript formatting checklist:

- MS Word document of text needed (including editable tables)
- MS Word document of supplemental text needed, if applicable (including figure legends and editable tables)
- Brief Statement describing supplementary information needed, if applicable (in subsection at end of Materials & Methods)
- Please include a data availability statement preceding the Acknowledgments section. Please see <https://rupress.org/jgp/pages/editorial-policies#data-availability-statement>
- Figures created at sufficient resolution and in acceptable format (including supplemental if applicable). If working in Illustrator, we prefer .ai or .eps file format. If working in Photoshop please use 600dpi/1000dpi .tiff or .psd file format. Minimum resolution at estimated print size: Minimum resolution for all figures is 600 dpi. For figures that contain both photographs and line art or text, 600 dpi is highly recommended. Figures containing only black and white elements (line art, no color, and no gray) should be 1,000 dpi. Maximum figure size is 7 in wide x 9 in high (17.5 x 22.8 cm) at the correct resolution. <https://jgp.rupress.org/fig-vid-guidelines>
- Supplemental figures, if any, conforming to same guidelines as manuscript figures (noted above)
- If images resemble one from a prior publications, the author must seek permissions (to reproduce or adapt) from the original publisher. [You can resubmit your paper while waiting to hear back from the original publisher but please keep us updated]
- All authors must complete a disclosure form prior to acceptance. A link to complete the form has been sent to all coauthors. Please provide the editorial office with updated email addresses if necessary

Reviewer #1 (Comments to the Authors):

The authors have addressed my previous criticisms and I have no further comments.

Reviewer #2 (Comments to the Authors):

The revision offers some improvements. Admittedly, addressing the underlying issues would require an entirely different model, which is not feasible within the scope of this paper. Given that this is a purely modeling study under the Hypothesis category, publication would be acceptable. I do, however, have a few points to raise:

1. The voltage sensor may move in a multi-step, biphasic manner; however, this should not be interpreted as two distinct, independent movements of the voltage sensor, each separately coupled to hyperpolarization- or depolarization-dependent gating. This key assumption was not supported by any experimental evidence. As authors stated in the response that "We agree that it has not been demonstrated whether these movements correspond to hyperpolarization- and depolarization-dependent gating, as no experiments have been conducted in bipolar mutants." This linear, sequential modeling is highly speculative.
2. The seven-state sequential model indeed involves eight free parameters in total, which is a relatively high number. In this type of modeling, equilibrium constants are treated as free parameters unless otherwise constrained. Therefore, it is not four free parameters as the authors stated in their response.
3. The sequential model, which assumes two voltage-sensor movements oppositely coupled to voltage, provides an overly simplified yet convenient framework for describing CNBD gating. There were advances from the earlier Zagotta-Hoshi-Aldrich (ZHA) model for shaker K channel to the more comprehensive Horrigan-Aldrich (HA) modular allosteric model for BK channels, which also considers voltage-independent pore opening and allosteric coupling factors between domains. Further work will be needed for the modeling of CNBD gating.

Reviewer #3 (Comments to the Authors):

I would like to thank the authors who carefully considered my suggestions. Nonetheless, I strongly recommend making a comparison with Hummert and colleagues (PLoS Comput Biol. 2018) where the voltage-independent pore-opening transition of HCNs was already proposed. Indeed, as the authors correctly stated in the rebuttal, namely "there is no definitive evidence that pore opening in HCN channels is completely voltage independent", this comparison will strengthen their model. Moreover, a comparison with Hummert and colleagues (PLoS Comput Biol. 2018) for me is also helpful to better elucidate a limit of the present gating model: The allosteric effects of binding/unbinding of cyclic nucleotides, which are known to allosterically modulate HCN and CNG channels. Such ligand-dependent effects are considered in the model published in PLoS Comput Biol. 2018.

Response to Reviewers:

Reviewer #1 (Comments to the Authors):

The authors have addressed my previous criticisms and I have no further comments.

Thank you for your constructive feedback and positive comments.

Reviewer #2 (Comments to the Authors):

The revision offers some improvements. Admittedly, addressing the underlying issues would require an entirely different model, which is not feasible within the scope of this paper. Given that this is a purely modeling study under the Hypothesis category, publication would be acceptable. I do, however, have a few points to raise:

1. The voltage sensor may move in a multi-step, biphasic manner; however, this should not be interpreted as two distinct, independent movements of the voltage sensor, each separately coupled to hyperpolarization- or depolarization-dependent gating. This key assumption was not supported by any experimental evidence. As authors stated in the response that "We agree that it has not been demonstrated whether these movements correspond to hyperpolarization- and depolarization-dependent gating, as no experiments have been conducted in bipolar mutants." This linear, sequential modeling is highly speculative.

We stated that the voltage sensor undergoes two distinct transitions (corresponding to three possible states), but we did not, by any means, imply that these transitions are independent of each other. Indeed, that would be impossible. If a voltage sensor is in a hyperpolarized open conformation, it cannot be in a depolarized open conformation at the same time.

An example of independent voltage-sensor transitions regulating channel gating is the HH model. Our model is nothing like that.

We respect your opinion that our model is “highly speculative” and encourage you to publicly challenge it so we can have an open debate and let others weigh in.

2. The seven-state sequential model indeed involves eight free parameters in total, which is a relatively high number. In this type of modeling, equilibrium constants are treated as free parameters unless otherwise constrained. Therefore, it is not four free parameters as the authors stated in their response.

In our response, we have shown that the 5-state model fits all the functional data when structural information is not taken into account. This was done because the model the reviewer was referring to also did not account for any structural constraints, especially the new data from the chimera. It has four free parameters, as we have shown.

3. *The sequential model, which assumes two voltage-sensor movements oppositely coupled to voltage, provides an overly simplified yet convenient framework for describing CNBD gating. There were advances from the earlier Zagotta-Hoshi-Aldrich (ZHA) model for shaker K channel to the more comprehensive Horrigan-Aldrich (HA) modular allosteric model for BK channels, which also considers voltage-independent pore opening and allosteric coupling factors between domains. Further work will be needed for the modeling of CNBD gating.*

Agreed!

Reviewer #3 (Comments to the Authors):

I would like to thank the authors who carefully considered my suggestions. Nonetheless, I strongly recommend making a comparison with Hummert and colleagues (PLoS Comput Biol. 2018) where the voltage-independent pore-opening transition of HCNs was already proposed. Indeed, as the authors correctly stated in the rebuttal, namely "there is no definitive evidence that pore opening in HCN channels is completely voltage independent", this comparison will strengthen their model.

Moreover, a comparison with Hummert and colleagues (PLoS Comput Biol. 2018) for me is also helpful to better elucidate a limit of the present gating model: The allosteric effects of binding/unbinding of cyclic nucleotides, which are known to allosterically modulate HCN and CNG channels. Such ligand-dependent effects are considered in the model published in PLoS Comput Biol. 2018.

Thank you for your feedback. We have added the following sentence (line 313) in the main text to bring this idea to the reader's attention.

“Hummert et al. (2018) proposed that the HCN channel voltage-sensor undergoes a two-step transition, followed by a voltage-independent pore-opening transition, primarily based on modeling ionic currents.”